# Heat stable and intrinsically sterile liquid protein formulations

Atip Lawanprasert [1], Harminder Singh [1], Sopida Pimcharoen[1], Mariangely González Vargas[1], Arshiya Dewan[2,3], Girish S. Kirimanjeswara [2,3,4,5] & Scott H. Medina [1,5] ✉

Over 80% of biologic drugs, and 90% of vaccines, require temperature-controlled conditions throughout the supply chain to minimize thermal inactivation and contamination. This cold chain is costly, requires stringent oversight, and is impractical in remote environments. Here, we report chemical dispersants that non-covalently solvate proteins within fluorous liquids to alter their thermodynamic equilibrium and reduce conformational flexibility. This generates non-aqueous, fluorine-based liquid protein formulations that biochemically rigidify protein structure to yield thermally stable biologics at extreme temperatures (up to 90 °C). These non-aqueous formulations are impervious to contamination by microorganismal pathogens, degradative enzymes, and environmental impurities, and display comparable pre-clinical pharmacokinetics and safety profiles to standard saline protein samples. As a result, we deliver a fluorochemical formulation paradigm that may limit the need for cold chain logistics of protein reagents and biopharmaceuticals.

Environmentally sensitive proteins, which include hormones, cytokines, enzymes, and antibodies, require temperature control throughout the supply chain to avoid thermal inactivation, enzymatic degradation, oxidative damage and microorganismal contamination[1–3]. Maintaining cold chain logistics is expensive (>$58 billion projected by 2026)[4], and failures can lead to significant patient harm. Several strategies have sought to address the thermal instability of proteins via sequence engineering[5,6], immobilization within synthetic scaffolds[7,8], and addition of molecular stabilizers[9,10]. Additionally, removing water to create lyophilized powders improves thermal stability as, in the dry state, proteins are forced to occupy a static structure with restricted polypeptide chain mobility[11]. Polymeric additives have attracted particular attention due to their ability to protect proteins during supercooling, ice crystallization, sublimation, and desorption processes[12,13]. Although these methods have achieved success, they are not broadly applicable across protein classes and must be empirically tailored to the specific biologic of interest, at considerable effort and cost. Further, removal of the solvation shell during lyophilization can lead to protein crowding and irreversible aggregation, making drying methods unsuitable for many proteins. In all cases, cold storage is required to maintain product sterility and minimize protein degradation.

Water solvent molecules are the medium through which these adverse unfolding, inactivation, and contamination processes occur. During thermal denaturation, for example, increased kinetic energy of water molecules disrupts the intramolecular hydrogen bonds, hydrophobic interactions, and Van der Waals forces that maintain the protein's folded state. This, combined with increasing conformational entropy of extended polypeptide chains, causes a loss of protein secondary, tertiary, and quaternary structure. Solvation of unfolded regions by water further promotes structural collapse and stabilizes the denatured, inactive state[14,15].

Herein, we explore the replacement of traditional aqueous solvents in protein formulations with a non-aqueous and nonpolar perfluorocarbon (PFC) liquid. Unlike water, PFCs rarely accept hydrogen

[1]Department of Biomedical Engineering, Pennsylvania State University, University Park, PA, USA. [2]Department of Veterinary and Biomedical Sciences, Pennsylvania State University, University Park, PA, USA. [3]Center for Molecular Immunology and Infectious Disease, Pennsylvania State University, University Park, PA, USA. [4]Center for Infectious Disease Dynamics, Pennsylvania State University, University Park, PA, USA. [5]Huck Institutes of the Life Sciences, Pennsylvania State University, University Park, PA, USA. ✉e-mail: shm126@psu.edu

bonds, due to their low polarizability, and are too bulky to readily penetrate the internal hydrophobic structures of proteins. Such a solvent restricts the conformational plasticity of proteins and subsequently alters their thermodynamic equilibrium. However, proteins are generally insoluble in PFC liquids. In order to overcome this immiscibility, we identified a privileged fluorochemical compound that promiscuously adsorbs to protein surfaces to provide a fluorophilic coating that enables the protein's efficient dispersion within non-aqueous PFC liquids, without disrupting the structure or function of the biologic. We show this strategy yields liquid protein formulations that remain stable and bioactive at temperatures up to 90 °C, while maintaining a solvation shell at the protein surface to avoid irreversible aggregation. Additionally, these non-aqueous samples cannot be contaminated by bacterial and fungal pathogens that require aqueous solvents for survival, and demonstrate an enhanced resistance to degradation by proteolytic enzymes and noxious oxidative compounds relative to aqueous controls.

## Results

### Protein dispersant characterization

We initially set out to identify a reagent that promiscuously binds to protein surfaces to create a generalizable coating strategy suitable for PFC dispersion (Fig. 1a). Our prior work showed that perfluorononanoic acid (PFNA) adsorbs to proteins via hydrogen bonding with solvent accessible backbone moieties and amino acid side chains (Fig. 1b), with an average PFNA:protein stoichiometry that varies from 1000:1 to -1700:1 depending on protein identity[16,17]. Interestingly, we previously observed PFNA-mediated conformational changes to decorated proteins even at sub-stoichiometric ratios, suggesting that PFNA may alter protein PFC solubility through multiligand ensemble effects rather than a de-facto 1:1 protein–ligand interaction[14]. To

further characterize these solubilizing effects, we examined the dispersion performance of PFNA using five model proteins: green fluorescent protein (GFP), bovine serum albumin (BSA), human hemoglobin (Hb), β-Galactosidase (β-Gal), and human serum immunoglobulin G (IgG). This group was selected as it represents a broad variety of protein classes, including carrier proteins, enzymes, and antibodies. Using perfluorohexane (PFH) as an exemplary fluorous solvent, solubilization assays demonstrated PFNA-mediated dispersion ranged from 69% to 100% efficiency, depending on protein identity (Fig. 1c). To assess the biophysical determinants of protein dispersion into PFH we used in silico protein-ligand docking to compare PFNA's dispersion efficiency with its predicted protein binding energies and solvent exposure (Fig. 1d–g). Interestingly, these analyses did not show a statistically significant correlation between PFNA dispersion performance and its docking energy with the protein surface (Fig. 1e). This suggests that PFNA binding avidity has little impact on dispersion efficiency of the biologic. Instead, the solvent-accessible and solvent-excluded surface areas (SASA Fig. 1f, and SESA, Fig. 1g, respectively) of bound PFNA were found to directly correlate to protein solubility in PFH. Mechanistically, this suggests that the contact area between PFNA's polar carboxylate and the protein surface, as well as the exposure of PFNA's fluorinated tail within the bulk PFC solvent, jointly determine the ligands' ability to partition coated biologics into the fluorous phase. These structure-performance relationships may be useful in predicting the dispersion efficiency of future, application-specific, protein candidates not tested here.

### Thermal stabilization of proteins via fluorous dispersion

Thermal analysis of fluorous-dispersed samples began using circular dichroism (CD) spectroscopy, an optical technique that monitors

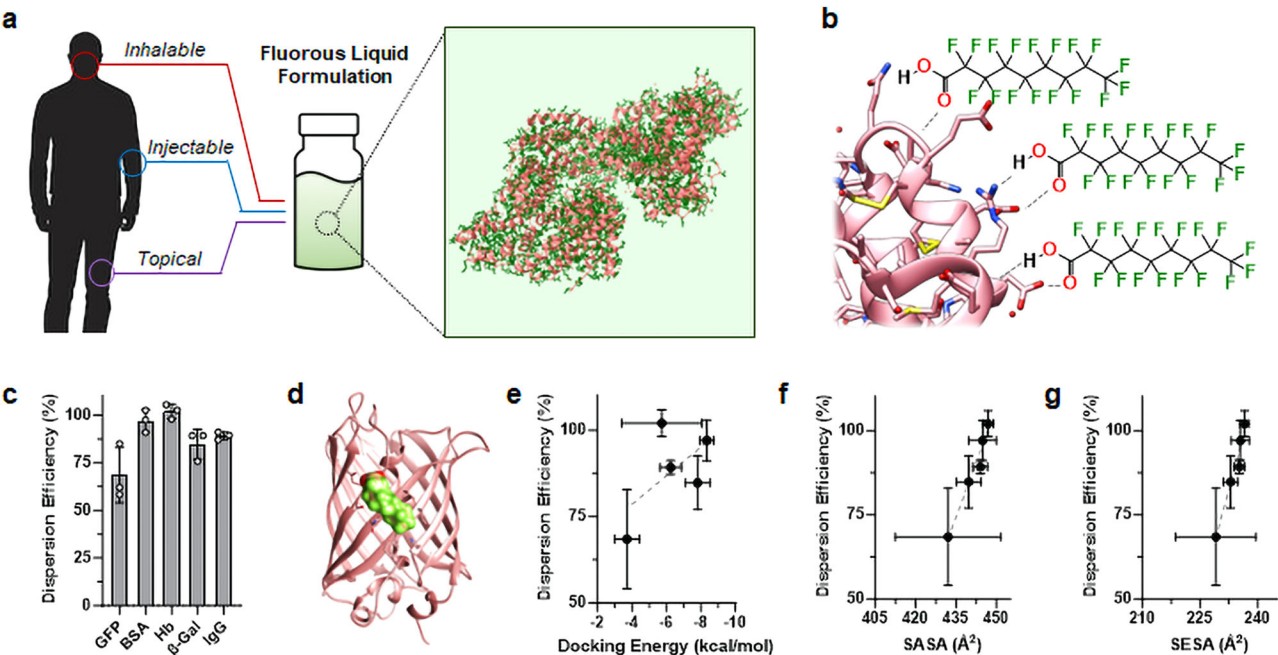

**Fig. 1 | Fluorous dispersion of proteins via PFNA coatings. a** Conceptual schematic showing an example protein structure (pink) coated by the fluorine-rich amphiphile PFNA (dark green) to enable its dispersion into non-aqueous PFCs liquids (light green background). This strategy yields thermally stable and intrinsically sterile protein formulations that we envision can be inhaled, injected, or topically applied. Human silhouette and vial graphic obtained from Michal Sanca/Shutterstock.com and PIXARTIST/Flaticon.com, respectively. **b** Graphical representation of hydrogen bonding (dashed line) between PFNA's carboxylate and the surface of an exemplary protein (pink). **c** Dispersion efficiency of green fluorescent protein (GFP), bovine serum albumin (BSA), Hemoglobin (Hb), β-Galactosidase

(β-Gal) or human serum immunoglobulin G (IgG) into perfluorohexane in the presence of the PFNA additive (PFNA:protein molar ratio of 1000:1). Results displayed as percent soluble protein relative to initial loading. **d** Representative result from MedusaDock 2.0 in silico protein-ligand docking showing adsorption of a single PFNA ligand (green/red) to the surface of GFP (pink). Correlation of protein dispersion efficiency to predicted docking energy (**e**), solvent accessible surface area (SASA, **f**) or solvent excluded surface area (SESA, **g**) of the PFNA ligand. Linear trends are shown by dashed lines, with $R^2$ values of 0.35, 0.96, and 0.95 for (**e**, **f**, and **g**), respectively. Data shown in (**c** and **e–g**) represent the average ± s.d. of $n = 3$ replicate analyses. Source data for panels c and e–g are provided as a Source Data file.

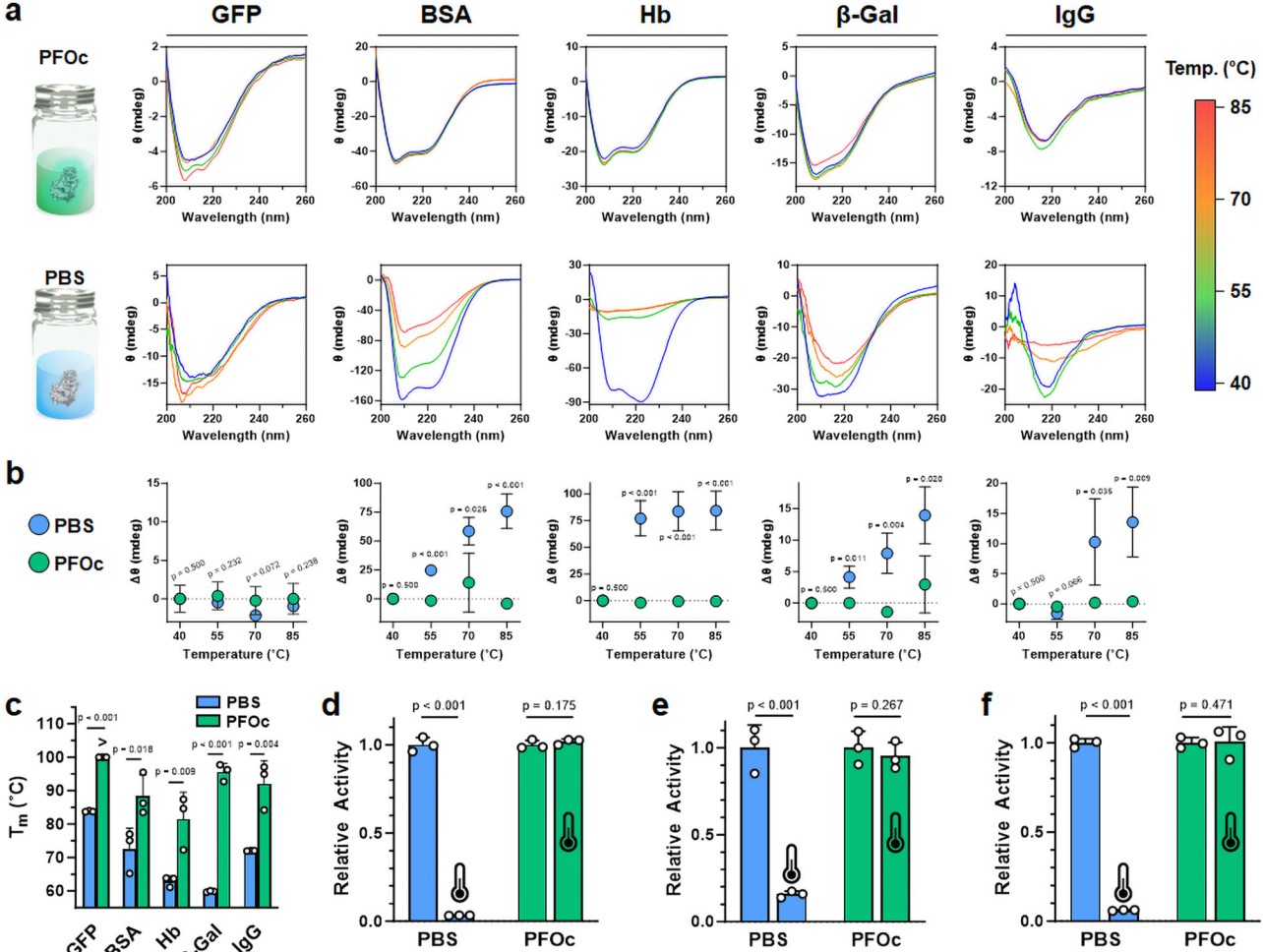

**Fig. 2 | Protein thermal stability. a** Temperature-dependent CD spectra of the indicated test protein solubilized in PFOc (*top row*) or PBS (*bottom row*). Vial graphic utilized in formulation concept images provided by Karon Arnold/Publicdomainpictures.net. **b** Change in minimum CD ellipticity for each test protein solubilized in PBS (blue) or PFOc (green) at the indicated temperature. Protein identity is identified by the in-register labels at the top of (**a**). **c** DSC measured melting temperature ($T_m$) for each of the indicated test proteins. ">" symbol indicates $T_m$ could not be reached at the highest achievable cell temperature of 100 °C.

Bioactivity of β-gal (**d**), GFP (**e**), and trypsin (**f**) in PBS (blue) or PFOc before (25 °C) and after incubation at 90 °C for 30 min. Heat-treated samples are indicated by the thermometer icon. Data shown in (**b–g**) represent the average ± s.d. of $n = 3$ technical replicates. Statistical significance between conditions is indicated by a line, or in the case of (**b**) is measured for PBS relative to PFOc at each temperature interval, using one-sided Student's *t*-test. Source data for panels a–f are provided as a Source Data file.

protein secondary structure. Initial studies found that PFH (bp = 56 °C) was not a compatible solvent as it evaporated before a melting temperature of the dispersed protein could be reached. We therefore used perfluorooctane (PFOc, bp = 103 °C), and demonstrated a minimal change in secondary structure when PFNA-dispersed proteins were heated to 85 °C (Fig. 2a). Conversely, saline control formulations showed denaturation of nearly all the proteins tested at this temperature, as exemplified by the significant loss of β-sheet (212 nm) and α-helix (208 and 222 nm) canonical signals (see bottom row in Fig. 2a). The notable exception was GFP, which is natively heat stable with a reported melting temperature of >80 °C[18]. To further quantitate these results, we plot the change in minimum CD ellipticity for each protein formulation as a function of temperature (Fig. 2b). These analyses show that, in saline, unfolding begins to occur between 55 and 70 °C for most of the protein candidates, while PFOc formulations demonstrate negligible changes in ellipticity at all tested temperatures. In order to support these findings, we performed differential scanning calorimetry (DSC) analyses for all five test proteins dissolved in PBS or PFOc (Fig. 2c, Supplementary Fig. 1). Here, again, we could not obtain a discrete melting temperature ($T_m$) for GFP in the PFOc solvent, as it

surpassed the 100 °C limit of the instrument before a $T_m$ could be reached. However, in all other cases, protein $T_m$ was 1.2–1.6 times higher in PFOc than it was in the saline control solvent.

While these results are encouraging, even slight changes in protein structure can have profound consequences on bioactivity. Therefore, we next performed bioactivity assays to test whether PFNA-dispersed PFOc protein formulations retain their functionality at elevated temperatures (Fig. 2d–f). These experiments required us to select proteins with an established functional assay, and so we, therefore, monitored β-gal conversion of a colorimetric substrate (Fig. 2d) and the intrinsic fluorescence of GFP (Fig. 2e) pre- (25 °C) and post (90 °C) heating for 30 min. As expected, β-gal and GFP in saline lost >85% of their bioactivity after heating, while no statistically significant change in activity of the PFOc dispersed proteins was detected under the same conditions. To further demonstrate the generalizability of this platform, we also prepared PFNA-dispersed samples of bovine trypsin in PFOc, and tested its heat-dependent bioactivity via a colorimetric substrate conversion assay (Fig. 2f). Like β-gal and GFP, trypsin maintained its enzymatic activity after being heated to 90 °C in PFOc, while saline controls were inactivated under the same conditions.

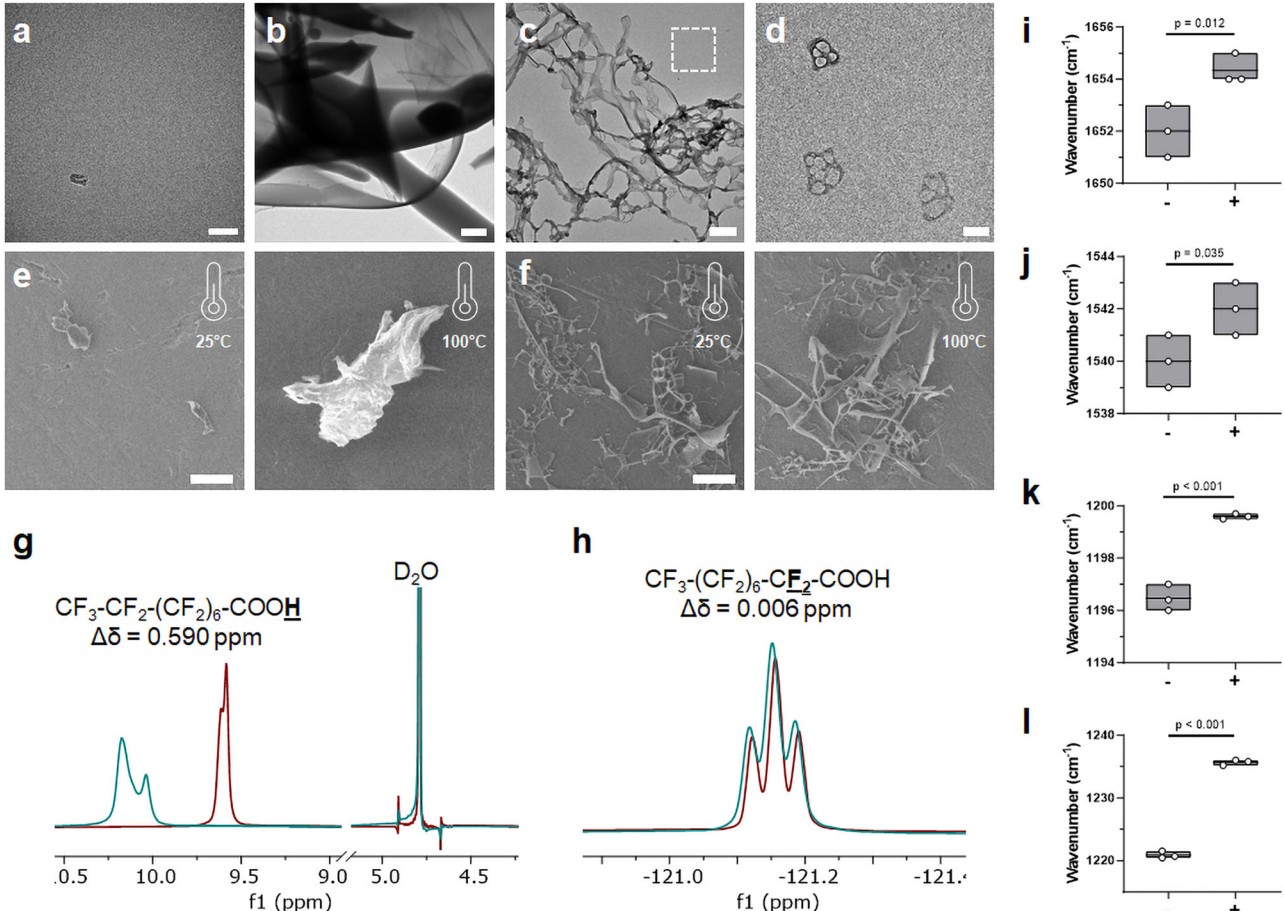

**Fig. 3 | Biophysical analysis of PFNA–protein interactions.** Representative transmission electron micrograph of monomeric BSA in saline (**a**) and amorphous protein aggregates (**b**) following unaided addition of the protein to PFOc. Scale bar in (**a**, **b**) represent 50 nm and 2 μm, respectively. **c** Representative transmission electron micrograph of PFNA-coated BSA assemblies in PFOc. Scale bar = 500 nm. **d** Magnified region of interest in (**c**) (white dashed box) showing individual PFNA-functionalized protein globules. Scale bar = 100 nm. Representative scanning electron micrographs of BSA in saline (**e**) or dispersed into PFOc via PFNA (**f**) before (25 °C) and after (100 °C) heating for 30 min. Scale bars = 50 μm. $^1$H (**g**) and $^{19}$F (**h**) NMR spectral regions of PFNA in PFOc before (maroon) and after (teal) coupling to BSA. Average wavenumber of BSA amide I (**i**) and amide II (**j**) bands before (−) and after (+) addition of PFNA. Average wavenumber of PFNA $CF_2$ (**i**) and $CF_3$ (**j**) bands before (−) and after (+) addition of BSA. Box plots shown in (**i**–**l**) represent data from $n = 3$ technical replicates, with line at mean and box bounds reflecting maxima and minima values. Statistical significance between conditions in panels i–l is indicated by a line using one-sided Student's *t*-test. Source data for panels i–l are provided as a Source Data file.

Next, we utilized a series of imaging and spectroscopy techniques to gain a deeper mechanistic understanding of how the PFNA additive and PFOc solvent work together to thermally stabilize dispersed proteins, using BSA as an exemplary candidate. Transmission electron micrographs shown in Fig. 3a-b demonstrate that, without the PFNA additive, BSA aggregates into amorphous flocculates in PFOc. When coated with PFNA, however, BSA assembles into a fibrillar network, where each fiber has a "beads-on-a-string" morphology (Fig. 3c). Magnification of the regions between the higher-ordered fibrillar mesh shows lower-order globules composed of PFNA-coated protein clusters (Fig. 3d). Together, this suggests that PFNA-coated proteins ("beads") are in equilibrium between incorporation within the fibrillar network ("string") and delocalization within the bulk PFOc solvent. The assembled structures likely sterically constrain the conformational dynamics of the incorporated protein and, thereby, restrict thermal unfolding. To further test this assertion, scanning electron microscopy (SEM) was used to probe the assembled state before and after heating (Fig. 3e,f). To calibrate our interpretation of these images, we began by collecting a baseline of heat-denatured BSA in sterile saline (Fig. 3e). As expected, BSA is monomeric at room temperature (25 °C) but readily denatures and aggregates after incubation at 100 °C for 30 min. Protein coated with PFNA and dissolved in PFOc, however, maintained their fibrillar

morphology after heat treatment (Fig. 3f), indicating these stabilizing structures likely represent a thermodynamic minimum state.

$^1$H (Fig. 3g) and $^{19}$F (Fig. 3h) NMR spectroscopy (see Supplementary Fig. 2 and 3 for full spectra) was next used to mechanistically interrogate the interaction of PFNA's carboxylic acid and fluorinated tail, respectively, with BSA. Proton NMR shows a 0.59 ppm downfield shift of PFNA's carboxy proton upon interaction with BSA, indicating the ligand adsorbs to the protein surface via hydrogen bonding. Conversely, a negligible change in the proximal $CF_2$ peak suggests PFNA's perfluorinated tail weakly interacts with the protein surface, instead more likely extended into PFOc to enable solvation by the bulk solution. Unfortunately, distal $CF_2$ and $CF_3$ groups could not be interrogated due to overlap of their $^{19}$F NMR signals with the PFOc solvent peaks. We therefore performed Fourier transform infrared (FTIR) spectroscopy to further probe changes to the BSA and PFNA structure upon coupling (Fig. 3i–l, see Supplementary Figs. 4 and 5 for full FTIR spectra). In the presence of the PFNA ligand, BSA's amide I (Fig. 3i) and amide II (Fig. 3j) bands show a blue shift of ~2 cm$^{-1}$ relative to the unfunctionalized protein. Prior studies of protein vibrational spectra indicate these types of band shifts may result from changes in the electrostatic environment and/or β-sheet and α-helical structure[19,20]. We have previously shown that, in

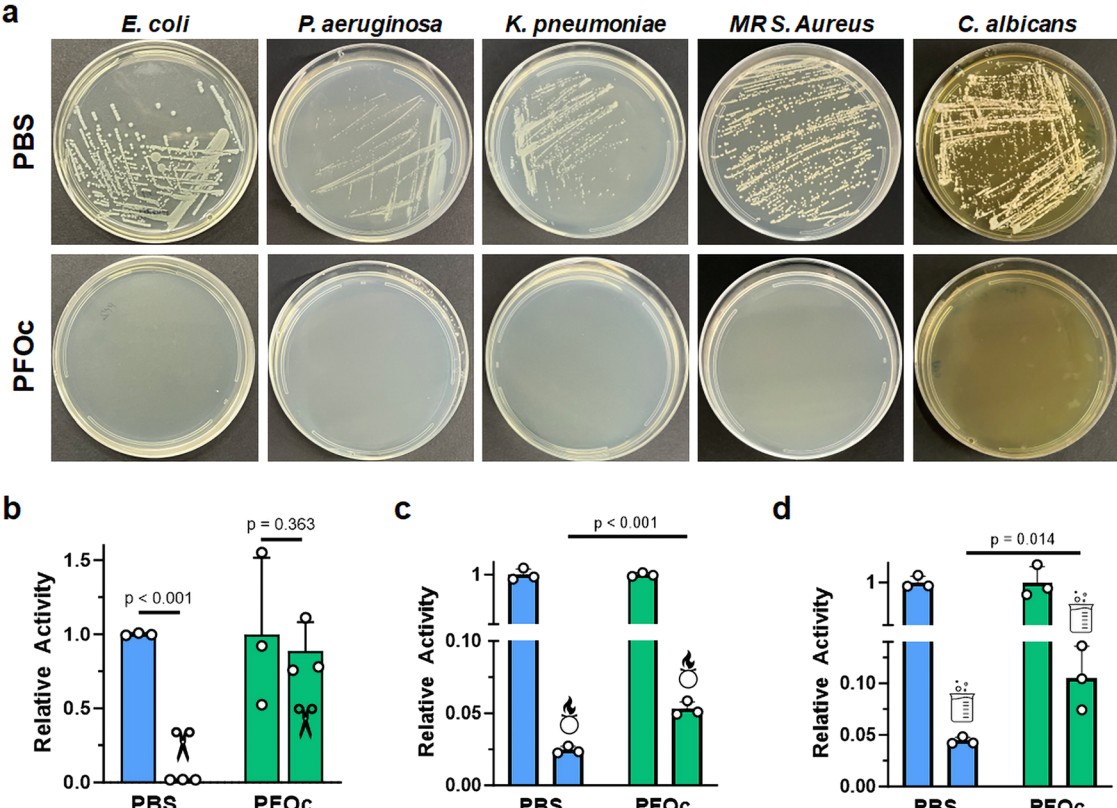

**Fig. 4 | Shelf sterility and stability of fluorous protein formulations.**
**a** Representative optical images of agar plates after inoculation with PBS (*top*) or PFOc (*bottom*) BSA samples contaminated with the indicated pathogen. MR = methicillin-resistant. Relative activity of β-Gal protein dispersed in PBS (blue) or PFOc (green) solvents without (no icon) and with (icon) contamination by proteinase K (**b**), bleach (**c**), or hydrochloric acid (**d**). Data are shown as average ± s.d. of $n = 3$ technical replicates. Statistical significance between conditions is indicated by a line using one-sided Student's *t*-test. Source data for panels b–d are provided as a Source Data file.

aqueous systems, PFNA adsorption can lead to a significant stabilization of protein β-sheet regions[16].

Complementary FTIR analyses of PFNA's fluorinated groups showed a 3 cm$^{-1}$ and 15 cm$^{-1}$ blue shift in CF$_2$ (Fig. 3k) and CF$_3$ (Fig. 3l) bands, respectively, upon interaction with BSA. Given that we observed a minimal change in the environment of PFNA's CF$_2$ group adjacent to the carboxylic acid by NMR (Fig. 3h), we interpreted these band shifts to indicate local aggregation of PFNA bound to the protein surface. This assembly is most likely driven by favored fluorine-fluorine interactions between adjacent ligands. We believe this assembled PFNA intermediary layer potentiates the formation of a PFOc hydration shell around the coated protein, thereby reducing its conformational dynamics and improving its resistance to thermal denaturation. Additional $^{19}$F NMR experiments identified a concentration-dependent chemical shift in the solvent -CF$_3$ peaks in the presence of dispersed β-Gal (Supplementary Fig. 6), further supporting the existence of a PFOc solvation shell at the protein surface.

**Shelf sterility and stability of fluorous protein formulations**
We hypothesized that removal of the water solvent should enable our fluorous protein formulations to resist contamination by bacterial and fungal contaminants that require aqueous environments to survive. To test this, we streaked a hypodermic needle across a lawn of each pathogen prepared on agar, and then submerged the contaminated needle into BSA protein formulations prepared in either PBS or PFOc solvents. To model the types of organisms that may be encountered in a healthcare setting, we tested the human bacterial pathogens *E. coli, P. aeruginosa, K. pneumoniae* and Methicillin-resistant *S. aureus* (MRSA), as well as the human fungal pathogen *C. albicans*. Contaminated liquid

formulations were then incubated at 37 °C overnight and replated onto agar plates to assess growth. As anticipated, BSA prepared in PBS was readily contaminated by all five pathogens, as indicated by the visible growth of viable colonies, while PFOc samples remained sterile for over a month (Fig. 4a, Supplementary Fig. 7). It is important to note that these studies employed pathogen concentrations well above what would be normally encountered in a healthcare setting. Nevertheless, PFOc formulations resisted contamination, possibly due to dehydration of the microorganisms upon introduction to the fluorous solvent.

In addition to bacterial and fungal pathogens, protein therapies can be exposed to inactivating environmental agents, including bacterial proteases, oxidizing cleaners, and acidic disinfectants. We experimentally modeled these conditions by adding an aliquot of proteinase K, a chlorine and sodium hydroxide mixture (e.g., bleach), or hydrochloric acid, respectively, to PFOc and PBS protein samples (Fig. 4b–d). These studies employed β-Gal as an exemplary biologic and substrate conversion assay used to evaluate protein activity after incubation with the denaturant. In the presence of proteinase K, β-Gal dissolved in PBS was completely inactivated, while there was no statistically significant change of protein activity in PFOc under similar conditions (Fig. 4b). A possible explanation for this is that proteinase K is coated by the PFNA additive after addition to the PFOc solvent, leading to its spatial segregation from the co-dispersed β-Gal protein. While fluorous protein formulations are impervious to degradation by proteases, they were found to be less resistant to chemical denaturants (Fig. 4c, d). Although the activity of β-Gal dispersed in PFOc, and then treated with an oxidizer (Fig. 4c) or acid (Fig. 4d), was statistically higher than that of the PBS controls, both conditions led to a ≥ 90% loss in functionality after only a few minutes of incubation. It is worth

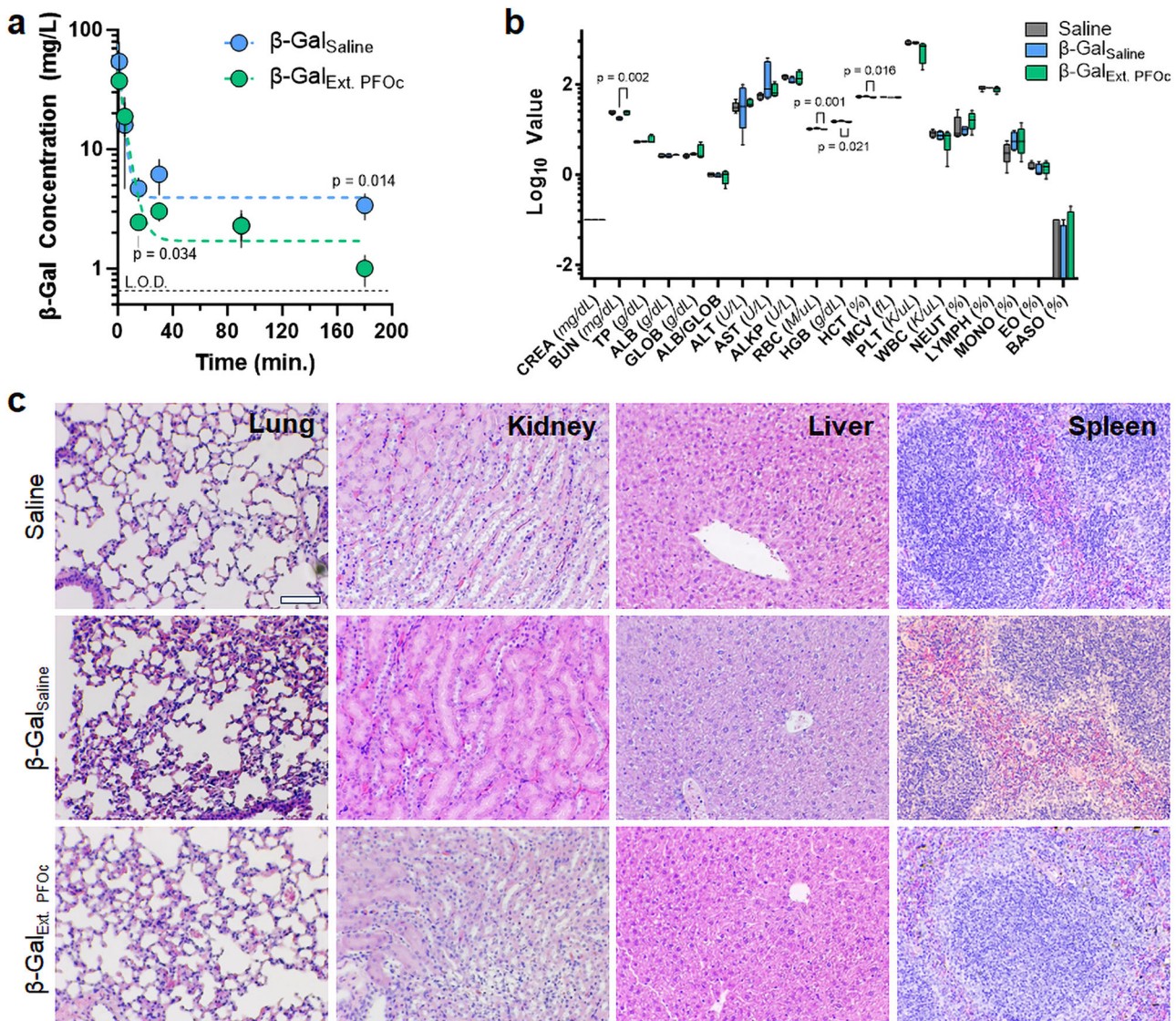

**Fig. 5 | Preliminary in vivo pharmacokinetics and acute toxicity. a** Time-dependent serum concentration of β-Gal delivered systemically in either saline (β-Gal_Saline, blue) or extracted PFOc (β-Gal_Ext. PFOc, green) vehicle. Data are shown as a semi-log plot of average ± s.d. of $n = 4$–5 technical replicates, with the exception of β-Gal_Saline at $t = 30$ min. which has $n = 4$ technical replicates due to a sampling error. The limit of detection (L.O.D.) is 0.66 mg/L (Supplementary Fig. 8, represented as dashed line on plot). Calculated pharmacokinetic parameters are reported in Supplementary Fig. 9. **b** Serologic toxicology results from C57BL/6J mice 24 h after administration of saline (control), β-Gal_Saline or β-Gal_Ext. PFOc. Data are shown as box and whisker plot ± s.d. of $n = 5$ technical replicates for saline (control) and $n = 4$ technical replicates for β-Gal_Saline and β-Gal_Ext. PFOc. Statistical significance in panels **a** and **b** determined using one-sided Student's $t$-test. For clarity, only statistically significant $p$ values are shown, all other comparisons resulted in $p > 0.05$. Full-size image and tabulated results can be found in Supplementary Fig. 11 and Supplementary Table S1, respectively. **c** Representative histopathologic images of lung, kidney, liver, and spleen tissue section from C57BL/6J mice 24 h after administration of saline (control), β-Gal_Saline, or β-Gal_Ext. PFOc. Each imaging group consisted of $n = 4$ mice, with four random fields per section collected at 10× magnification in a blinded manner. Scale bar = 100 μm. Full-size image can be found in Supplementary Fig. 12. Source data for panels a and b are provided as a Source Data file.

noting that denaturant concentrations employed here are significantly higher than what would typically be encountered in a healthcare setting. In a typical incidental exposure, protein therapies would be contaminated by noxious vapors, rather than directly spiked with a solution of the denaturant as done in our experiments. Nevertheless, the improvement in stability for our PFOc protein samples highlights the potential of this platform in these applications and encourages further development.

### In vivo pharmacokinetics and toxicity
A final technologic milestone is to evaluate differences in the time-dependent serum concentration of coated proteins, as well as assess the toxicity of fluorous formulations relative to saline controls. To demonstrate that PFOc dispersion does not alter the circulatory half-life of proteins we intravenously administered β-Gal formulations to C57BL/6 mice and monitored protein serum concentrations over time (Fig. 5a). Due to the large solution volumes given (100 μL, ~10% of mouse blood volume), we elected to first extract the PFOc dispersed proteins into sterile saline before injection to avoid hyponatremia (see methods; extraction was quantitative). Unfortunately, attempts to reduce the injection volume to circumvent this limitation resulted in too low of a delivered protein concentration to be detected by our substrate conversion assay. Despite this additional formulation step, we envision that future studies in large animals and humans will allow for direct injection of the fluorous dispersion, without pre-extraction, given the feasibility of reducing infusion

volume ratios and the non-toxic nature of most perfluorocarbon solvents[21]. Results in Fig. 5a show that the serum half-life of β-Gal delivered from the PFOc dispersion ($t_{1/2}$ = 5.80 min) is similar to native β-Gal administered in saline ($t_{1/2}$ = 7.12 min.). It is important to note that half-life is calculated in our studies via regression analysis of results from a fluorescent substrate conversion assay, with subsequent translation of this bioactivity readout to protein concentration via a calibration curve (see methods and Supplementary Fig. 8). Additional pharmacokinetic parameters were calculated for β-Gal$_{Saline}$ and β-Gal$_{Ext. PFOc}$ formulations (Supplementary Fig. 9) and results were found to be statistically similar ($p > 0.05$). Follow-up proteolysis studies demonstrated that β-Gal remains active in mouse serum for >6 h (Supplementary Fig. 10), supporting conclusions that its short in vivo serum half-life is due to rapid tissue distribution and/or renal elimination, rather than enzymatic degradation. This corroborates prior murine studies which found that β-Gal is cleared from serum in <15 min[22,23]. In sum, our data suggest that fluorous dispersion does not change the enzymatic function or pharmacokinetic properties of functionalized proteins.

Finally, to determine any acute toxic effects of the residual fluorous dispersant, blood chemistry and histology were performed 24 h after treating mice with β-Gal delivered from

saline or PFOc extractions. Serological renal, hepatic, and hematologic toxicology screens showed statistically significant changes in blood urea nitrogen, red blood cell count, hemoglobin, and hematocrit between β-Gal$_{Ext. PFOc}$ and β-Gal$_{Saline}$ (Fig. 5b, Supplementary Fig. 11 and Table S1). However, none of these markers were statistically different between PFOc β-Gal and the sham saline injection control, suggesting these small changes are not clinically meaningful. This is further corroborated by histologic organ analyses that did not identify signs of necrosis, cellular infiltration, inflammation, or hemorrhage in lung, kidney, liver, and spleen tissue excised from treated animals (Fig. 5c, Supplementary Fig. 12). Collectively, these results suggest the fluorous media used in our protein dispersion formulations are unlikely to induce acute toxic side effects. This corroborates a body of clinical evidence showing that intravenously injected perfluorocarbon solvents are cleared from the lungs during respiration to avoid adverse effects due to long-term tissue bioaccumulation[24].

## Discussion

Scalable strategies to limit the cold chain dependence of protein biopharmaceuticals rely on co-solvent additives (e.g., glycerol) or freeze-drying. In both cases, the goal is to drive the protein into a more compact, and thermally stable, state. However, there are several limitations inherent to current state-of-the-art formulation methodologies, including the need to empirically optimize additive conditions for each protein candidate, the propensity for many biologics to irreversibly aggregate during drying, and the liability of contamination at each stage of the cold chain network. Liquid fluorous formulations hold promise to comprehensively address these barriers by presenting an indiscriminate dispersion methodology with little-to-no requirements for protein-specific optimization, maintains a bulk solvent medium to avoid protein desolvation, and is intrinsically impervious to contamination due to its non-aqueous nature. We envision that translation of this formulation technology to other proteins can be accomplished by simply mixing a lyophilized sample, or concentrated aqueous fraction, with a PFC solvent containing PFNA to generate the fluorous dispersion. The identity of the PFC solvent can be rationally chosen to control the viscosity and boiling point, depending on application-specific needs. In our hands, we observed that use of lyophilized starting material, and agitation of solutions via a rotisserie mixer, yield higher dispersion efficiencies relative to concentrated aqueous samples and vortex mixing, respectively. Our mechanistic studies show the assembled

dispersant complex at the surface of the protein thermally stabilizes the biologic without compromising its structure and biologic function. Animal studies further demonstrate this approach does not alter the serum half-life and safety profile of the dispersed proteins. Yet, further development of this formulation paradigm is necessary to reduce the barriers to its practical implementation. Of foremost priority is to maximize the soluble concentration of protein within the fluorous phase to enable direct injection of the therapeutic without compendial extraction or processing. Additionally, it would be beneficial to develop methods to disperse aqueous protein samples into the stabilizing fluorous media, rather than relying on lyophilized or concentrated products. Such a goal may be realized by developing amphiphilic dispersants that undergo hierarchical assembly at the water-fluorous interface, thereby creating nanoscale receptacles that bind to proteins and mediate an aqueous-to-perfluorocarbon exchange.

These advances may be realized by building upon prior work in non-aqueous protein technologies. For example, our formulation approach is a contemporary alternative to hydrophobic ion pairing (HIP) methods reported several decades ago for protein dissolution in organic solvents[25,26]. In HIP strategies, the replacement of polar counterions at the protein surface with anionic surfactants creates a hydrophobic coating that transitions the biologic's solubility from aqueous systems to nonpolar organic solvents. Meyer et al., for example, demonstrated that the enzyme α-chymotrypsin could be stably dispersed within alkane and chlorocarbon solvents when modified with the detergent sodium bis(2-octyl)sulfouccinate[25]. Like the work reported here, this strategy retained the globular structure of the protein and significantly enhanced its thermal stability. Similar results were reported for peptides complexed with sodium dodecyl sulfate[26]. This work came out of a broader interest in nonaqueous enzymology, where substrate biotransformations could be conducted in organic solvents containing little or no water, as pioneered by Klibanov, Russell, Dordick, and others[27–30]. The efficiency of HIP, however, is limited by the availability of basic residues on the solvent-accessible protein surface to electrostatically interact with anionic detergents. Fluorous dispersion, presented here, offers a promiscuity advantage in that it exploits general hydrogen bonding between the dispersant and the protein backbone to mediate solubilization in nonaqueous solvents, rather than electrostatic interactions. As a result, with further development, this methodology may open a diverse area of formulation technologies that yield thermally stable and intrinsically sterile protein biopharmaceuticals.

## Methods

### Protein dispersion

Solubilizing proteins in PHF or PFOc was performed following a previously developed protocol[16], with minor modification. In brief, 1 mM PFNA dissolved in the fluorous solvent was added to lyophilized protein stocks in a 1.5 mL centrifuge tube to achieve a final concentration of 10 μM for BSA, GFP, and Hb, 5 μM for IgG, and 1 μM for β-Gal. To ensure proper mixing, each sample was parafilmed and then sequentially vortexed (30 s), sonicated (1 minute), and again vortexed (30 s). Samples were centrifuged for 5 min at 1950 × $g$ to remove insoluble aggregates. To quantify percentage of dispersed protein, 100 μL of the supernatant was transferred into each well of 96-well plate and solvent allowed to evaporate at 37 °C. Dried protein residue was resuspended in 100 μl 1X PBS with equal parts Coomassie blue reagent for Bradford Assay. Samples were shaken at 25 °C for 10 min and read for absorbance at 595 nm using a plate reader (Biotek Cytation 3). Protein concentration was calculated based on calibration curves generated in PBS. Protein in PBS and PFH were used as positive and negative controls, respectively, at the appropriate concentration. PBS was used as a blank. Dispersion efficiency (%) was calculated from protein

concentration measurements using the following equation:

$$Dispersion\ Efficiency(\%) = \frac{([Sample] - [Negative\ Control])}{([Positive\ Control] - [Negative\ Control])} x\ 100\%$$

(1)

## Molecular docking

COACH-D[31] molecular docking and consensus algorithm was used to predict docking site and PFNA ligand docking thermodynamic properties. In brief, protein structures obtained from the PDB (GFP: 3UFZ; BSA: 4F5S; Hb: 6FQF; β-Gal: 3VDB; IgG: 2VUO) were imported and five pockets for PFNA ligand binding were generated based on minimum energy. PFNA ligand was retrieved from Zinc Database with the ID: 38141429. Docking energy was computed for each rotamer-pocket combination. SASA and solvent-excluded surface area (SESA) of the bound PFNA ligand were calculated for each prediction using UCSF Chimera[32].

## Spectroscopy

CD spectroscopy was performed on proteins prepared in PBS, or dissolved in PFOc using 1 mM PFNA, with concentrations based on the previously established quantities in the protein dispersion experiments. Samples were then heated to a designated temperature between 40 °C and 85 °C for 15 min. Before analysis, PFOc dispersed samples were extracted into room temperature PBS buffer for detection, as the fluorous solvent caused solvent-dependent dichroic aberrations. The extraction of protein was quantitative, as determined by Bradford assay. Sample measurements were taken at the indicated temperature using a Jasco J-1500 Circular Dichroism Spectrometer (Easton, MD). Replicates ($n = 3$) were performed for each condition with representative spectra reported.

Nuclear Magnetic Resonance (NMR) spectroscopy was performed by adding a 60 mM PFNA in PFOc solution to lyophilized BSA to achieve a final 1000:1 PFNA:BSA molar ratio. Samples were then transferred to thin wall precision tube (Wilmad-LabGlass; Vineland, NJ), with Norell Coaxial inserts containing $D_2O$ as the locking solvent. NMR spectra were collected on a Bruker NEO-400, equipped with a double resonance broadband observe iProbe (capable of automatic tuning and matching) for $^1$H and $^{19}$F nuclei observation. All spectra were recorded by using zg30 (1D sequence with 30° flip angle for $^1$H NMR consisting of 32 scans, sweep width = 20.4850 ppm; origin point = 6.175 ppm) and zgig (inverse gated-decoupling 1D pulse sequence for $^{19}$F NMR consisting of 512 scans, sweep width = 241.4836 ppm; origin point = −100.0 ppm) in the Bruker library at 298 K. Data was analyzed with Mnova software. Interrogation of a fluorous solvation shell was examined using $^{19}$F NMR spectroscopy by titrating dispersions containing fixed concentration of PFNA (1 mM) with varying concentrations of β-Gal (0–100 μM) in PFOc. $^{19}$F signal was referenced relative to 2-(Trifluoromethyl)acrylic acid.

Fourier Transform Infrared (FTIR) spectrometry was performed on protein samples prepared by mixing an equivalent volume of BSA (20 μM) and PFNA (2 mM) in PBS with each other and allowing the sample to incubate at 37 °C for 1 h. Samples were then frozen at −80 °C and lyophilized overnight before analysis using a Bruker VERTEX 70 FTIR spectrophotometer (Billerica, MA) outfitted with an LN-MCT detector (4000 – 800 cm$^{-1}$, backward output). Replicates ($n = 3$) were performed for each condition with representative spectra reported.

## Differential scanning calorimetry

Solutions of proteins (10 μM) prepared in PBS or PFOc (1 mM PFNA) were added to sample holders appropriate for each instrument. A Malvern MicroCal VP-capillary DSC (Malvern, PA) was employed for aqueous samples. TA Instruments DSC Q2000 (New Castle, DE) was utilized for PFOc samples. Different instruments were required for

these assays due to solvent incompatibility issues. Samples were subjected to a heat ramp at 2 °C/min from 40 °C to 100 °C during which heat capacity ($C_P$) was measured following normalization to the reference cell containing blank solvent. Melting temperatures were determined as the peak of the $C_P$ curve.

## Colorimetric conversion assay

β-Gal and Trypsin were diluted in PBS or dispersed in PFOc, using 1 mM PFNA, at 1 μM and 10 μM concentrations, respectively. Samples were then either incubated at room temperature (25 °C) or subjected to elevated temperature (90 °C) for 30 min, before cooling samples back to room temperature over another 30-min interval. For PFOc samples, proteins were extracted into PBS before analysis as both enzymatic assays require an aqueous environment to be operational. This was accomplished by adding an equal volume of the buffer to the PFOc sample, vortexing for 15 s, and then removing the PBS supernatant containing protein samples for analysis. The extraction of proteins into PBS is quantitative. An equal volume of PBS containing 4 mg/mL ONPG for β-Gal or 1 mg/mL BAEE for trypsin, was added to the appropriate sample and incubated following manufacturers' instructions. Substrate conversion was then measured at 420 nm or 400 nm for ONPG and BAEE, respectively, using a BioTek Cytation 3 microplate reader (Winooski, VT). Relative activity was calculated by normalizing test sample data to the enzymatic activity of proteins at room temperature in their respective solvent environment.

## Electron microscopy

A 5 μL aliquot of BSA (10 μM) prepared in PFOc using PFNA (1 mM) was added protein to a copper grid, dried overnight, and TEM imaging performed at 200 kV using a FEI Tecnai LaB6 electron microscope (Hillsboro, OR). BSA (1 μM) diluted in water was included as a control. SEM imaging was performed following a similar protocol, with the modification that heat-treated samples were incubated at 100 °C for 1 h. After samples had cooled to room temperature, 10 μL of each sample was added to a stub and allowed to dry overnight. Residues were coated with Au/Pd and imaged at 10 kV using a Zeiss SIGMA VP-SEM with VPSE G3 detector (Dublin, CA).

## Contamination assays

For pathogen contamination experiments, *E. coli* (101-1), *P. aeruginosa* (PAO1), *K. pneumoniae* (NCTC 9633), and Methicillin-resistant *S. aureus* (MRSA; USA300) were cultured in MHB broth. C. *albicans* (3147) cultured in YPD broth at 37 °C. Pathogens were plated on MHB (*E. coli, P. aeruginosa, K. pneumoniae*, MR *S. aureus*) or YPD (*C. albicans*) agar to develop lawns. All broth cultures were grown at 37 °C in a shaking incubator (200 rpm, and plates were cultured in a static incubator, as advised by the Clinical and Laboratory Standards Institute (CLSI). To contaminate PBS or PFOc liquid samples, a 21 g needle was streaked in a single pass across the microbial lawn, and dipped into liquid samples containing BSA (10 μM). Samples were then incubated at 37 °C for up to four weeks before streaking onto an agar plate. Plates were incubated at 37 °C overnight before qualitatively visualizing colony formation.

Enzymatic degradation was investigated by preparing solutions of β-Gal (1 μM) and proteinase K (10 μM) in PBS or dispersed in PFOc (1 mM PFNA). An equal volume of each solution was mixed to induce degradation, and the sample was incubated at 37 °C for 24 h. After incubation, PFOc samples were extracted into PBS for bioactivity determination by adding an equal volume of PBS and vortexing for 15 s. An equal volume of PBS containing 4 mg/mL ONPG was added to each sample, incubated for 15 min at 37 °C, and absorbance measured at 420 nm using a BioTek Cytation 3 microplate reader (Winooski, VT).

Environmental degradation was performed by preparing a solution of β-Gal (1 μM) in PBS or dispersed in PFOc (1 mM PFNA). An aliquot of 1% v/v of 10% bleach or 4% v/v of 0.1 M HCl was added to each sample to assess oxidation or acid-mediated denaturation,

respectively. After 30 s of incubation at 37 °C, an equal volume of PBS containing 4 mg/mL ONPG substrate was added to the sample and incubated for 3–5 min to allow for complete substrate conversion. Absorbance was measured at 420 nm using a BioTek Cytation 3 microplate reader (Winooski, VT), and compared to blank buffer, or a stock β-Gal (1 μM) solution, as negative and positive controls, respectively, to calculate relative activity.

## Animal experiments

Murine studies were performed under approved IACUC protocol 202101978, utilizing 6-week-old female C57BL/6 mice. Mice were housed in shared cages with sterile bedding, with a 12 h day night cycle and room maintained at 72 °F ± 1° and humidity 30–70%. Sex was not considered as a variable in this study to minimize variance during pilot pharmacokinetic studies. Two groups ($n = 4–5$) received 100 μL of 1 μM solutions of β-Gal either prepared in sterile PBS, or dissolved in PFOc (1 mM PFNA) an extracted into sterile PBS. Samples were administered by tail vein injection to C57BL/6 J mice, with a final dose of 1.55 mg/kg for a 30 g mouse. At 1, 5, 15, 30, 90 and 180 min post administration, mice were sacrificed, and blood samples collected via cardiac puncture. Concentration of β-Gal in serum was determined by conversion of the fluorescent substrate, 4-Methylumbelliferyl-α-D-galactopyranoside (μ-GAL). In brief, the collected whole blood in a serum tube was centrifuged at 2000 × $g$ for 4 min to extract serum containing β-Gal protein. 20 μL of the serum was then mixed with 180 μL μ-GAL (final substrate concentration of 1 μM). Samples were quickly pipette mixed and incubated in dark at room temperature for 5 min. Fluorescent intensity was measured with $\lambda_{ex} = 360$ nm and $\lambda_{em} = 440$ nm using a BioTek Cytation 3 microplate reader (Winooski, VT). The background fluorescent intensity from serum was determined from serum dilution in PBS without μ-GAL substrate, and protein concentration was calculated relative to a calibration curve. Serum stability was tested by incubating β-Gal (1 μM) in mouse blood serum ($n = 3$), before removing 20 μL aliquots at different time points and performing the μ-GAL assay as previously described in the reported methods above. Samples were diluted 10× in PBS before measurements, with data reported in relative fluorescence units (r.f.u.).

To assess acute toxicity, animals were administered i.v. solutions of β-Gal either prepared in sterile PBS, or dissolved in PFOc (1 mM PFNA) and extracted into sterile PBS, as described above. After 24 h mice were sacrificed, and blood was collected via cardiac puncture. A minimum of 100 μL of whole blood was added to EDTA for complete blood count. The remainder was centrifuged (1000 × $g$, 10 min) to isolate serum for chemistry analysis. Serological analyses were performed by the Pennsylvania State University Animal Resources Program. Tissues (lungs, livers, kidneys, and spleens) from euthanized animals were collected and stored in 10% buffered formalin immediately after euthanasia. Four 5 μM sections from each organ were mounted onto glass slides and stained with Hematoxylin and Eosin (H&E). Four random fields in each section were examined under a microscope at 5× and 40× magnification. Tissue-relevant clinical features were compared across samples. Specifically, lungs were examined for infiltration of poly and mononuclear cells, signs of hemorrhage, and perivascular infiltration. Kidneys were examined for signs of necrosis, cellular infiltration, and hemorrhage. Livers were examined for hepatic cell necrosis, inflammation, and hemorrhage. Spleens were examined for changes in white and red pulp structure, as well as signs of abnormal cellular infiltration.

## Statistical analysis

Unless stated otherwise in the manuscript, data represents $n \geq 3$ technical replicates and presented as mean ± standard deviation. Significance between groups was determined by one-sided Student's $t$-test, with $p$ values reported in the appropriate plot.

## Software

Data collection was performed using Verios G4 xT and Mnova. Data analysis was performed using GraphPad Prism 9, Microsoft Excel version 2410, and ImageJ version 1.54k.

## Reporting summary

Further information on research design is available in the Nature Portfolio Reporting Summary linked to this article.

## Data availability

The processed Supplementary data are available at https://doi.org/10.6084/m9.figshare.27932079. All data supporting the findings of the study are available from corresponding author upon request. Source data are provided with this paper.

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

## Acknowledgements

Transmission and scanning electron microscopy were performed at the Penn State Microscopy and Cytometry Facility, University Park, PA. NMR spectroscopy was performed at the Penn State NMR Facilities, University Park, PA. We acknowledge the Huck Institute X-ray Crystallography Facility for use of the CD spectrophotometer and Differential Scanning Calorimetry instrumentation (NIH grant S10-OD025145). We would like to thank Penn State Materials Research Institute for access to Fourier-Transform Infrared Spectrometry instrumentation and Differential Scanning Calorimetry. We would also like to thank Drs. Arun Sharma and Asif Raza for their guidance in the calculation and interpretation of pharmacokinetic parameters. Molecular graphics and analyses were performed with UCSF Chimera, developed by the Resource for Biocomputing, Visualization, and Informatics at the University of California, San Francisco, with support from NIH P41-GM103311. Funding for this work was provided by NSF DMR–1845053, NIH 1R35-GM142902, and DARPA D21AP10182 to S.H.M.

## Author contributions

A.L. and S.H.M. conceived this work and wrote the manuscript. A.L., H.S., and S.P. performed and analyzed most experiments. H.S. additionally performed dispersion screens, $^{19}$F NMR, and $^{1}$H NMR. M.G.V. performed computational studies. A.D. and G.S.K. oversaw in vivo studies. Correspondence and requests for materials should be addressed to Scott H. Medina (shm126@psu.edu).

## Competing interests

The authors declare no competing interests.
