## [Transparent Peer Review file · Nature Communications]

Heat Stable and Intrinsically Sterile Liquid Protein Formulations

Corresponding Author: Professor Scott Medina

Version 0:

Reviewer comments:

Reviewer #1

(Remarks to the Author)

Review of

The manuscript by Lawanprasert et al. describes the formulation of proteins in perfluorinated solvents. I feel like the work is well done, but needs some context and for the authors to address some specific issues, as listed below.

The interaction between PFNA and the proteins is reminiscent of hydrophobic ion pairing (HIP), a phenomenon that was first described more than 30 years ago. The ability to dissolve proteins in nonaqueous solvents using HIP depends on the change in SASA, much like what is discussed lines 76-83. Moreover, HIP complexes display very high T_m values in these environments, much like in this study. Therefore, this literature must be presented and discussed at this juncture. I would recommend the following articles as starting points: Meyer et al., *Biopolymers* 1995, 35, 451-456 and Powers et al., *Biopolymers* 1993, 33, 927-932. I believe there are literature examples as well of enzyme retaining activity in completely nonaqueous solvents, most notably from Klibanov, Lenhoff, and others. These should also be included.

This is probably a minor point, but on line 93, the authors mention minima in the CD spectra. These are not minima per se. They are negative maxima. The bands reflect an intensity related to conformation (and the amount of alpha-helical structure), but just happen to have negative ellipticity.

Line 200. Not clear what is meant by "extract". Is this a dilution step into saline? If so, please state as such. This could be an issue for practical use, if there needs to be the use of a diluent. I am also concerned that there is some impact on blood chemistry in the animal studies. Any additional information on toxicity of perfluorous drug carriers would be appreciated.

In the references, many of the journal titles are not capitalized throughout (some are, many are not). In addition, there are two mistakes in the citations.

Ref 9. The author's name is O'Fágáin, C.

Ref. 11. The year is 2002 not 2020.

Reviewer #2

(Remarks to the Author)

This reviewer's comments are according to the instructions by the editor limited to the section entitled "In vivo equivalency and toxicity" and the related "Methods" sections:

The authors intend to establish 'bioequivalence' of their fluoruous protein formulation relative to a conventional protein formulation in saline by intravenous injection of a single dose on mice. While they claim that their formulation is non-toxic, they still first extract the beta-Gal in their fluoruous formulation into sterile saline solution before injecting it into the animal. This largely defies the purpose of the claimed 'bioequivalence' assessment. It remains unclear why they want to give this large volume of 150 mL which they are concerned will result in hyponatremia rather than reducing the dose and thus

volume for this assessment.

In addition they intent to show 'bioequivalence' which is based on the pharmacokinetic parameters area-under-the concentration-time curve and peak concentration. Instead of reporting these measures and/or the primary pharmacokinetic parameters clearance and volume of distribution, they report only half-life, and then claim that the fluorous formulation does not change the pharmacokinetics of beta-Gal. That is a completely unsupported conclusion based on reporting half-life alone.

Further, in the beginning of the paragraph, the authors mention that they want to demonstrate 'Pharmacokinetics and pharmacodynamics of proteins' are not influenced by their formulation. However, they only show pharmacokinetic data. The quantification of the beta-Gal 'bioavailability and bioactivity' is not performed by a chromatographic assay, mass spectrometry based techniques, or a ligand binding assay as usually applied for therapeutic proteins, but by a colorimetric assay based on beta-Gal's enzymatic activity. This is a highly inadequate technique to claim 'bioequivalence' or no change in pharmacokinetics between the formulations as the decline in Fig.5 could be the result of infinite combinations of pharmacokinetic elimination of beta-Gal concentrations and reduction of beta-Gal enzymatic activity in mouse serum. As such, the assay technique is utterly inadequate to show 'bioequivalence'.

Figure 5 b is unreadable due to its small size and the logarithmic scale. It would be more helpful to present the underlying data in table format in the supplemental material.

Reviewer #3

(Remarks to the Author)

The paper 'Heat Stable and Intrinsically Sterile Liquid Protein Formulations' by Lawanprasert and coworkers presents research findings on chemical dispersants that non-covalently solvate proteins within fluorous liquids to alter their thermodynamic equilibrium and reduce conformational flexibility, thereby enabling generation of non-aqueous protein formulations that show resistance to thermal degradation and microbiological contamination. The work presented here as well as the idea of developing stable formulations are of interest to biotech community. The authors also noted that the developed fluorochemical formulations may limit, or altogether eliminate, the need for cold chain logistics of protein reagents and biopharmaceuticals, which is a stretch and not supported by their findings (see comments below). Additional comments are included below for the authors to consider. Overall, the concept is interesting, and the initial data are indeed promising. It deserves further exploration.

Include additional clarification on how versatile the dispersant/solvent combinations are to justify statement "by presenting an indiscriminate dispersion methodology with little-to-no requirements for protein specific optimization". For example, consider including a few lines on what specifically need to be done for new proteins that are recombinantly produced as aqueous solution, say a monoclonal antibody or a cytokine, to generate a 100 mg/mL formulation.

Shelf Sterility and Stability of Fluorous Protein Formulations – This section demonstrates stability against microbiological contamination, which is one of the numerous instability/degradation pathways a therapeutic will face. And some of these destabilizing events can be slow and hence the need to establish a true shelf life over a long period (approximately 18-36 mo). The paper does not address the time component – one of the most important factors in determining shelf life.

Stability - A drug product faces many different stresses during production, handling, storage, administration etc (see many of these stress factors described in recent literature articles), which includes interfacial stresses. Discuss if the nonaqueous formulations are known/expected to provide benefits resisting these destabilizing stresses. Also, comment on volatility of PFH (and any impact on long term stability including loss of PFH/PFNA).

In vivo Equivalency and Toxicity – The authors elected to first extract the PFOc dispersed proteins into sterile saline before injection. This is a major concern when attempting to understand bioequivalency and toxicity. The data do not establish safety/PK unambiguously in the manner the study is conducted. Demonstration of safety of the EXACT (neat) formulation (i.e., not extracted/alterd version of the formulation) to be administered is equally important as the safety of the therapeutic protein itself. Excipients/stabilizers/additives can compromise safety including inducing local tolerance issues, distribution kinetics of the active, and adverse impact to blood/serum components. The references are not adequate to discuss safety in humans/animals, as the paper emphasizes applicability to protein therapeutics (i.e., for humans/animals).

The authors discussed difficulty of administering large solution volumes. What is the protein concentration in the solution administered? Discuss why the concentration couldn't be increased to enable smaller injection volume of the exact (neat) formulation? What is the highest protein concentration achieved in the PFNA/PFH combo?

Protein dispersion – this procedure starts with lyophilized protein – in contrast to the claims made by authors that this technology is an alternative to lyophilization etc. Discuss how this method would work with standard protein production processes that result in aqueous protein formulations.

Data in figure 2/3 versus their previous report (reference 14) that PFNA induces non-native secondary structure. Discuss the state of structure in the non-aqueous formulation (i.e., the actual formulation to be administered). It is important to understand the protein structure in the non-aqueous drug product formulation as well as what happens to the protein once it is administered.

The characterization assays are not performed on the representative formulations, and the starting point of samples in each assay uses different compositions/preps. Discuss the gap of not understanding structure of the proteins in the intended therapeutic formulation (to be administered as injected/inhaled as depicted in Fig. 1).

How complete is the extraction of protein into PBS from PFOc? If not 100%, discuss if only the fraction of active/native structure is extracted in PBS, leaving other fractions behind, and as a result skewing the relative activity assay data.

Without addressing these experimental gaps and gathering adequate data (at the minimum, discussing the gaps), it is highly speculative conclusion that they delivered "a first-in-class fluorochemical formulation paradigm that may limit, or altogether

eliminate, the need for cold chain logistics of protein reagents and biopharmaceuticals.”

Version 1:

Reviewer comments:

Reviewer #1

(Remarks to the Author)

Review of Lawanprasert et al.

Thank you for the hard work on revisions to the manuscript. I just have two important technical questions and a couple grammatical issues to address.

The authors describe that the PFNA ligand “promiscuously adsorbs to protein surfaces”. It would be most helpful to describe this in terms of stoichiometry. I would have expected the interaction to be more like hydrophobic ion pairing rather than simple adsorption to the surface via hydrogen bonding. A description of the amount of PFNA that is bound is an important detail to convey.

The authors state that the proteins maintain a “solvation shell at the protein surface to avoid irreversible aggregation”. Can you please provide evidence (spectroscopic, water content, etc.) that this is true? Indirectly, it appears to be correct based on the structural assessment of the proteins, but this is a critical point and strong statement and it needs to be justified, especially since they also talk about water removal in these systems.

Overall, the authors do a good job in this revised manuscript to demonstrate the potential utility of such dispersed proteins. Biophysical details on the method of preparation and characteristics of the dispersed materials is equally important, in my opinion.

In two places, sentences start with “To”. These should really start with “In order to...”

In the last paragraph, “it’s” should be “its”

Reviewer #2

(Remarks to the Author)

The authors have tried to diligently address the comments raised by this reviewer. Nevertheless, the section entitled ‘In vivo equivalency and toxicity’ is still substandard. It is highly recommended that the authors consult with a scientist with extensive pharmacokinetic experience and expertise to bring this section to a commonly acceptable level.

Besides general improvements, the following critical issues still need to be addressed:

In line 196, the authors claim that they want to detect differences in bioavailability. This is done in pharmacokinetics by comparing area-under-the-plasma-concentration time curves (AUC). Comparisons of half-life are utterly insufficient (and blatantly wrong) to achieve this goal.

In line 199, the authors claim that they monitor ‘serum bioavailability over time’. Again, an utterly incorrect pharmacokinetic concept. What they likely do is monitor serum concentration levels (or enzymatic activity levels) over time.

Line 200-203: The need for the extraction procedure still remains to be explained. The authors did that in the response to the reviewer comments, but not in the manuscript.

Line 205-208: While highlighting the lack of substantial differences in half-life, this is utterly insufficient for a pharmacokinetic comparison. Other pharmacokinetic parameters need to be compared as well (AUC, CL, Vd etc.)

Line 208-209: The authors claim that ‘Half -life is measured in our studies via a fluorescent substrate conversion assay...’. Again, this is scientific nonsense: Functional enzyme activity as a surrogate for active protein concentration is measured. Half-life is derived from those enzyme activity assay measurements over time via some type of regression analysis. This imprecision of description and wording of the authors in their scientific methodologies is disturbing, as it is likely not only limited to the pharmacokinetic assessments.

The toxicology assessment in lines 214-218 is largely meaningless as it is done with the extracted protein rather than the PFOc dispersed protein. As the latter is the goal to be administered to animals and humans, toxicity assessments should be focused on those formulations to assess their feasibility for translation into humans.

Reviewer #3

(Remarks to the Author)

The revised article, 460388_1, Heat Stable and Intrinsically Sterile Liquid Protein Formulations, by Lawanprasert and coworkers provided adequate responses to most queries including additional data (thank you). However, two concerns remain (see below) regarding the practical utility of the technology/formulation presented here. But this reviewer agrees that the technology/process/formulation presented here serve as proof of principles for improved thermal stability and shelf sterility, acknowledging the lack of demonstration on bioequivalency, toxicity, administration of adequate dose etc. For example, with the possibility of improvements in protein partitioning and solubility, as noted by the authors in their response, one can imagine the dose regimen getting closer to reality (at least for some proteins).

Concerns are:

(A) Extraction of the therapeutic

Extraction of the therapeutic by end user prior to administration is a high hurdle for injectables (that requires aseptic handling, preparation including transfer, and administration – the authors can review recent literature on these topics for challenges and controversies, as well as difficulty of implementing compendial and regulatory guidelines in addition to State-specific policies). Adding such a step defeats the purpose of improving access to drugs. The authors also noted that their process works better for lyophilized protein than aqueous formulation (aqueous formulation is the form produced for nearly all protein-based injectables, prior to converting to drug product). Thinking of the entire process for protein PFOc, (a) addition of lyophilization process, (b) addition of 1 or more custom devices for extraction/transfer/administration, and (c) the need for conducting the extraction by the end user, collectively makes the manufacturing cost and logistics high barriers. It should be noted that a lyophilized drug product already provides vastly improved thermal stability including long-term shelf stability relative to aqueous formulations (as well as slowing down potential microbial growth). The authors are advised to consult, if needed, with biotech professionals who have experience in commercializing biologics drugs in global markets.

(B) Lack of toxicity & bioequivalency data, lack of ability to produce adequate dose

The additional discussions on safety (lack of toxicity) and bioactivity/half-life are appreciated, however the lack of relevant experiments/data, as noted by this and other reviewers remains a gap. This is acknowledged by the authors in the revised manuscript.

A new gap emerged. With the added information, the achievable protein concentration appears to be orders of magnitude lower than the therapeutic IgG human doses (typically hundreds of mg total dose, and IgG concentration often exceeding 50 or 100 mg/mL in the recent years). So, the delivered dose via this new formulation will have to improve vastly, and ideally the process starting with aqueous formulation instead of requiring to lyophilize first. The authors are encouraged to add a few lines of discussion on why the presence of water (aq formulation) makes partitioning inefficient/inadequate, and if there is a way to circumvent it (without having to lyophilize).

Version 2:

Reviewer comments:

Reviewer #1

(Remarks to the Author)

The authors have made great efforts to improve the quality of the manuscript, which I appreciate. I think the paper is ready to be published except for one issue.

On lines 61-64, the authors claim that the stoichiometry is 1731. I find this to be strange. First, one should speak to a molar ratio (for example, 1731:1). First, how do determine this value to four significant figures? Second, this will vary from protein. Third, this value is not supported by the two cited references. They speak to systems where the ratios are 1000:1 to 1300:1. This needs to be corrected.

Reviewer #3

(Remarks to the Author)

Thanks for trying to address my queries. I have no additional comments.

Reviewer #4

(Remarks to the Author)

I have now assessed the response to Reviewer 2 and read the in vivo animal experimental description, results and interpretation.

Although this is quite interesting method, there are multiple limitations and I would list the things that need to be addressed to improve the presentation and increase the validity of the in vivo studies:

1. Type of b-Gal with its molecular weight needs to be indicated
2. The dose given needs to be presented also in mg/kg to make the comparison to other studies feasible
3. Exact blood sampling times need to be investigated.

4. The precision and accuracy data on the bioanalytical method needs to be presented. This is the key for any PK study
5. Dedicated PK analysis should be performed with reporting key systemic PK parameters (Cl, Vd) as it was already indicated by Reviewer 2. This is not difficult and would enhance the work. AUC calculation only by GraphPadPrism sounds a bit strange...
6. Fig. 5: In vivo half-life and safety of fluorouracil protein formulations title needs to be changed to: In vivo serum PK. I am not sure that B and C figures present "safety" data. Ideally it should be a bit toned-down and reflect what has been exactly done as this is not a dedicated safety study.
7. In Fig. 5 A should be presented in semilogarithmic scale as this is a the common way of presenting the data in PK.
8. Ideally the stability of b-gal in blood, in vitro should be reported as well as it seems that the protein may undergo extensive systemic proteolysis which may impact also bioanalysis.
9. It would be very valuable if authors could give a reference to other studies reporting the PK parameters for b-Gal, especially when presenting the t_{1/2}.

It should be possible to address those questions rather easily. In addition, the authors should clearly highlight the limitations of existing studies and the need of further investigations.

Version 3:

Reviewer comments:

Reviewer #4

(Remarks to the Author)

Authors addressed all the questions. I do not have any additional comments.

Reviewer #1 The manuscript by Lawanprasert et al. describes the formulation of proteins in perfluorinated solvents. I feel like the work is well done, but needs some context and for the authors to address some specific issues, as listed below.	Author Response to Reviewer #1 We thank the reviewer for their careful review of our manuscript. Based on the comments provided we have made significant changes to the revised text. This has resulted in a significantly stronger manuscript overall. Details of these changes are described below.
1. The interaction between PFNA and the proteins is reminiscent of hydrophobic ion pairing (HIP), a phenomenon that was first described more than 30 years ago. The ability to dissolve proteins in nonaqueous solvents using HIP depends on the change in SASA, much like what is discussed lines 76-83. Moreover, HIP complexes display very high T_m values in these environments, much like in this study. Therefore, this literature must be presented and discussed at this juncture. I would recommend the following articles as starting points: Meyer et al., Biopolymers 1995, 35, 451-456 and Powers et al., Biopolymers 1993, 33, 927-932. I believe there are literature examples as well of enzyme retaining activity in completely nonaqueous solvents, most notably from Klibanov, Lenhoff, and others. These should also be included.	1. We thank the referee for emphasizing this important body of work. We have now included a discussion of HIP in the conclusions, and present similarities and contrasts of this methodology with our work. Seminal references from the HIP and the nonaqueous enzymology fields have been included. Addition of this discussion provides a previously missing contextual basis for our findings, and we believe produces a much stronger manuscript overall. We sincerely appreciate the suggestion from the reviewer.
2. This is probably a minor point, but on line 93, the authors mention minima in the CD spectra. These are not minima per se. They are negative maxima. The bands reflect an intensity related to conformation (and the amount of alpha-helical structure), but just happen to have negative ellipticity.	2. To avoid confusion we now refer to these CD features as 'β-sheet (212 nm) and α-helix (208 & 222 nm) canonical signals'
3. Line 200. Not clear what is meant by "extract". Is this a dilution step into saline? If so, please state as such. This could be an issue for practical use, if there needs to be	3. To prepare samples for animal studies, β-Gal was separated from PFOc via an aqueous solvent exchange method. This was done due to concerns from our animal team that infusion of a large volume of the

the use of a diluent. I am also concerned that there is some impact on blood chemistry in the animal studies. Any additional information on toxicity of perfluorous drug carriers would be appreciated.

immiscible PFC diluent (150 μ L, ~15% of mouse blood volume) would cause embolization and/or hyponatremia. Unfortunately, attempts to reduce the injection volume to circumvent this resulted in too low of a delivered protein concentration to be detected by our substrate conversion assay. We therefore settled on first extracting the PFOc dispersed protein into sterile saline before injection to obtain approval for our studies.

That said, in new, and yet unpublished work, we have developed 2nd generation dispersants that increase the soluble concentration of protein in PFOc by ~5 fold. We believe this will allow us to significantly reduce injection volumes and address this current limitation. However, this work is still on-going and, we feel, beyond the scope of this initial publication, where our goal is to establish proof of principle.

Yet, should we find that aqueous extraction is still necessary despite the improved dispersant design, we do not believe this represents an insurmountable barrier to future practical application of our technology. This is because extraction can easily be accomplished by adding an equal volume of saline to the PFOc suspension, and vortexing the solution for 30 seconds (extraction is quantitative). Although this adds an extra step, the thermal stability and shelf sterility of our formulation, we believe, still represent a substantive advance in protein storage.

Finally, regarding the toxicity of our fluoruous compounds. Several perfluorocarbon liquids, many of which are similar in structure to PFOc used here, are already FDA approved for use as ultrasonography contrast agents, blood oxygenation devices and ophthalmologic surgery reagents (Holman et al. Front. Chem. 2021; 9: 810029). Clinical studies show these compounds deposit in the liver shortly after injection, but due to poor metabolism re-enter the blood, bind to serum lipids, distribute to the lungs, and are ultimately cleared via respiration. The toxicity of our dispersive agent, PFNA, in humans is not known, which is why we conducted toxicity studies in mice. While there were statistically significant

	changes in some functional markers between β-Gal delivered from saline versus PFOc extractions, these markers were not statistically different when comparing PFOc extracted β-Gal to the sham saline injection control. This suggests that the fluoros media used in our protein dispersion formulations are unlikely to induce acute toxic side effects, however additional studies are certainly needed to add confidence to these findings before future clinical testing. All these valuable discussion points and clarifications have now been added to the revised manuscript, in both the results, discussion and conclusion sections.
--	--

4. In the references, many of the journal titles are not capitalized throughout (some are, many are not). In addition, there are two mistakes in the citations. Ref 9. The author's name is O'Fágáin, C. Ref. 11. The year is 2002 not 2020.	4. We thank the reviewer for catching these citation errors. They have been corrected, along with appropriate capitalization of all journal titles, in the revised manuscript references.
--	---

Reviewer #2

This reviewer's comments are according to the instructions by the editor limited to the section entitled "In vivo equivalency and toxicity" and the related "Methods" sections:

Author Response to Reviewer #2

We thank the reviewer for their targeted review of our previously titled section: *In vivo equivalency and toxicity*. Based on the comments provided we have made significant changes to the revised text, including re-titling this section to improve clarity. Details of these changes are described below.

1. The authors intend to establish 'bioequivalence' of their fluorous protein formulation relative to a conventional protein formulation in saline by intravenous injection of a single dose on mice. While they claim that their formulation is non-toxic, they still first extract the beta-Gal in their fluorous formulation into sterile saline solution before injecting it into the animal. This largely defies the purpose of the claimed 'bioequivalence' assessment. It remains unclear why they want to give this large volume of 150 mcL which they are concerned will result in hyponatremia rather than reducing the dose and thus volume for this assessment.

1. Extraction of PFOc dispersed β -Gal into saline for animal studies was done due to concerns from our animal team that infusion of a large volume of the immiscible PFC diluent (150 μ L, ~15% of mouse blood volume) would cause embolization and/or hyponatremia. Unfortunately, attempts to reduce the injection volume to circumvent this concern resulted in too low of a delivered protein concentration to be detected by our substrate conversion assay. We therefore settled on first extracting the PFOc dispersed protein into sterile saline before injection to obtain approval for our studies.

That said, in new, and yet unpublished work, we have developed 2nd generation dispersants that increase the soluble concentration of protein in PFOc by ~5 fold. We believe this will allow us to significantly reduce injection volumes and address this current limitation. However, this work is still on-going and, we feel, beyond the scope of this initial publication, where our goal is to establish proof of principle.

Yet should we find that aqueous extraction is still necessary despite the improved dispersant design, we do not believe this represents an insurmountable barrier to future practical application of our technology. This is because extraction can easily be accomplished by adding an equal volume of saline to the PFOc suspension, and vortexing the solution for 30 seconds (extraction is quantitative). Although this adds an extra step, the thermal stability and shelf sterility of

	our formulation, we believe, still represent a substantive advance in protein storage. That said, we agree with the referee that our studies do not meet the threshold of demonstrating bioequivalence, and so we have removed this term from the revised manuscript.
2. In addition they intent to show 'bioequivalence' which is based on the pharmacokinetic parameters area-under-the concentration-time curve and peak concentration. Instead of reporting these measures and/or the primary pharmacokinetic parameters clearance and volume of distribution, they report only half-life, and then claim that the fluorous formulation does not change the pharmacokinetics of beta-Gal. That is a completely unsupported conclusion based on reporting half-life alone. Further, in the beginning of the paragraph, the authors mention that they want to demonstrate 'Pharmacokinetics and pharmacodynamics of proteins' are not influenced by their formulation. However, they only show pharmacokinetic data.	2. We agree with the reviewer that use of the term bioequivalence was an overstatement of our experimental design and results. As mentioned above, we now remove use of this term. Accordingly, we have narrowed our discussion to indicate our experiments are focused on functional protein serum bioavailability and toxicity. We have additionally modified the conclusions statement to indicate that future pre-clinical PK/PD studies are needed before further translation of the technology is warranted.
3. The quantification of the beta-Gal 'bioavailability and bioactivity' is not performed by a chromatographic assay, mass spectrometry based techniques, or a ligand binding assay as usually applied for therapeutic proteins, but by a colorimetric assay based on beta-Gal's enzymatic activity. This is a highly inadequate technique to claim 'bioequivalence' or no change in pharmacokinetics between the formulations as the decline in Fig.5 could be the result of infinite combinations of pharmacokinetic elimination of beta-Gal concentrations and reduction of beta-Gal enzymatic activity in mouse serum. As such, the assay technique is utterly inadequate to show 'bioequivalence.	3. We chose a substrate conversion assay to specifically monitor the serum concentration of functional protein. Our intention was to simultaneously evaluate half-life and demonstrate that PFOc dispersion did not compromise enzymatic bioactivity of the dispersed protein. To clarify this point, we have now modified the title of this sub-section to read: In vivo Functional Half-Life and Toxicity. In addition, we have added the following statement: "It is important to note that half-life is measured in our studies via a fluorescent substrate conversion assay to evaluate functional changes in enzyme activity as a result of fluorous dispersion. It does not report on total protein content. Nevertheless, analogous functional half-lives of β-Gal_{Ext. PFOc} and β-Gal_{Saline} suggest that fluorous dispersion does not change the enzymatic

function or circulatory kinetics of the complexed protein.”

4. Figure 5 b is unreadable due to its small size and the logarithmic scale. It would be more helpful to present the underlying data in table format in the supplemental material.

4. As requested, we now included a full size figure (Supplementary Fig. 7) and tabulated results (Supplementary Table S1) in the supplemental material.

Reviewer #3

The paper 'Heat Stable and Intrinsically Sterile Liquid Protein Formulations' by Lawanprasert and coworkers presents research findings on chemical dispersants that non-covalently solvate proteins within fluorinated liquids to alter their thermodynamic equilibrium and reduce conformational flexibility, thereby enabling generation of non-aqueous protein formulations that show resistance to thermal degradation and microbiological contamination. The work presented here as well as the idea of developing stable formulations are of interest to biotech community. The authors also noted that the developed fluorochemical formulations may limit, or altogether eliminate, the need for cold chain logistics of protein reagents and biopharmaceuticals, which is a stretch and not supported by their findings (see comments below). Additional comments are included below for the authors to consider. Overall, the concept is interesting, and the initial data are indeed promising. It deserves further exploration.

1. Include additional clarification on how versatile the dispersant/solvent combinations are to justify statement "by presenting an indiscriminate dispersion methodology with little-to-no requirements for protein specific optimization". For example, consider including a few lines on what specifically need to be done for new proteins that are recombinantly produced as aqueous solution, say a monoclonal antibody or a cytokine, to generate a 100 mg/mL formulation.

Author Response to Reviewer #3

We appreciate the thoughtful comments from the reviewer. Based on the concerns raised we have performed additional experiments and made significant changes to the manuscript text. We believe this produced a stronger manuscript overall. Details of these changes are discussed below.

1. As requested, we now discuss the application of our formulation strategy to other proteins in the revised manuscript conclusion. Specifically, we have added the following text:

"We envision that translation of this formulation technology to other proteins can be accomplished by simply mixing a lyophilized sample, or concentrated aqueous fraction, with a PFC solvent containing PFNA to generate the fluorinated dispersion. The identity of the PFC solvent can be rationally chosen to control the viscosity and boiling point, depending on application-specific needs. In our hands, we observed that use of lyophilized starting material, and agitation of solutions via a rotisserie mixer, yields higher dispersion efficiencies relative to concentrated aqueous samples and vortex mixing, respectively. Our mechanistic studies show the assembled dispersant complex at the

	surface of the protein thermally stabilizes the biologic without compromising its structure and biologic function. Animal studies further demonstrate this approach does not alter the serum half-life and safety profile of the dispersed proteins, warranting future pre-clinical pharmacokinetic and pharmacodynamic studies.”
2. Shelf Sterility and Stability of Fluorous Protein Formulations – This section demonstrates stability against microbiological contamination, which is one of the numerous instability/degradation pathways a therapeutic will face. And some of these destabilizing events can be slow and hence the need to establish a true shelf life over a long period (approximately 18-36 mo). The paper does not address the time component – one of the most important factors in determining shelf life.	2. Based on the referee’s suggestion we have now performed one-month sterility studies. Here, PFOc dispersed protein samples were contaminated with the bacterial pathogens P. aeruginosa or Methicillin-resistant S. aureus and allowed to incubate for four weeks, with weekly plating of samples to evaluate contamination. These new results are shown in Supplementary Fig. 6, and reproduced below for convenience. The plating images show that PFOc samples remain uncontaminated for up to 4 weeks. Unfortunately, because of the revision deadline, we did not have time to carry out our experiments beyond this one month time period.

Supplementary Fig. 6: Representative optical images of agar plates after addition of PFOc-BSA samples contaminated with (*top row*) *P. aeruginosa* or (*bottom row*) Methicillin resistant *S. aureus* and allowed to culture for 1 – 4 weeks.

3. Stability - A drug product faces many different stresses during production, handling, storage, administration etc (see many of these stress factors described in recent literature articles), which includes interfacial stresses. Discuss if the nonaqueous formulations are known/expected to provide benefits resisting these destabilizing stresses. Also, comment on volatility of PFH (and any impact on long term stability including loss of PFH/PFNA).

3. Outside of the thermal, chemical and enzymatic stresses tested in our studies, we did not specifically evaluate the influence of mechanical/handling factors on the stability of our fluoros protein formulations. That said, to prepare protein dispersions we continuously rotisserie mix the PFOc-protein samples overnight. Substrate conversion assays (Fig. 4b-d) show similar relative bioactivity between fluoros samples and saline controls, suggesting that mechanical forces exerted on the protein during PFOc dispersion does not negatively impact it's stability. In fact, PFOc has a kinematic viscosity that is ~15% higher than that of water ($V_{PFOc} = 1.03 \text{ mm}^2/\text{s}$, $V_{water} = 0.89 \text{ mm}^2/\text{s}$), suggesting that fluoros dispersions may marginally insulate dispersed protein from mechanical denaturation compared to aqueous counterparts. However, because we did not specifically test this condition in our experiments this is only speculation.

Finally, early in our studies we switched from using the perfluorocarbon solvent perfluorohexane (PFH) to perfluorooctane (PFOc). The high boiling point of PFOc (bp = 103°C) allowed a larger thermal range to be studied compared to PFH (bp = 56°C). Because PFOc has a boiling point like that of water, its evaporation rate was observed to be qualitatively similar to aqueous samples.

4. In vivo Equivalency and Toxicity – The authors elected to first extract the PFOc dispersed proteins into sterile saline before injection. This is a major concern when attempting to understand bioequivalency and toxicity. The data do not establish safety/PK unambiguously in the manner the study is conducted. Demonstration of safety of the EXACT (neat) formulation (i.e., not extracted/alterd version of the formulation) to be administered is equally important as the safety of the therapeutic protein itself. Excipients/stabilizers/additives can compromise safety including inducing local tolerance issues, distribution kinetics of the active, and adverse impact to blood/serum components. The references are not adequate to discuss safety in

4. We understand the reviewers' concern about the translation of our formulations given the described extraction step. This was done due to concerns from our animal team that infusion of a large volume of the immiscible PFC diluent (150µL, ~15% of mouse blood volume) would cause embolization and/or hyponatremia. Unfortunately, attempts to reduce the injection volume to circumvent this resulted in too low of a delivered protein concentration to be detected by our substrate conversion assay. We therefore settled on first extracting the PFOc dispersed protein into sterile saline before injection to obtain approval for our studies.

That said, in new, and yet unpublished work, we have developed 2nd generation dispersants

humans/animals, as the paper emphasizes applicability to protein therapeutics (i.e., for humans/animals).

that increase the soluble concentration of protein in PFOc by ~5 fold. We believe this will allow us to significantly reduce injection volumes and address this current limitation and intend to test this assertion in due course. However, this work is still on-going and, we feel, beyond the scope of this initial publication, where our goal is to establish proof of principle.

Yet should we find that aqueous extraction is still necessary despite the improved dispersant design, we do not believe this represents an insurmountable barrier to future practical application of our technology. This is because extraction can easily be accomplished by adding an equal volume of saline to the PFOc suspension, and vortexing the solution for 30 seconds (extraction is quantitative). Although this adds an extra step, the thermal stability and shelf sterility of our formulation, we believe, still represent a substantive advance in protein storage.

Finally, regarding the toxicity of our fluorous compounds. Several perfluorocarbon liquids, many of which are similar in structure to PFOc used here, are already FDA approved for human use as ultrasonography contrast agents, blood oxygenation devices and ophthalmologic surgery reagents (Holman et al. *Front. Chem.* 2021; 9: 810029). Clinical studies show these compounds deposit in the liver shortly after injection, but due to poor metabolism re-enter the blood, bind to serum lipids, distribute to the lungs and are ultimately cleared via respiration. The toxicity of our dispersive agent, PFNA, in humans is not known, which is why we conducted toxicity studies in mice. While there were statistically significant changes in some functional markers between β -Gal delivered from saline versus PFOc extractions, these markers were not statistically different when comparing PFOc extracted β -Gal to the sham saline injection control. This suggests that the fluorous media used in our protein dispersion formulations are unlikely to induce acute toxic side effects, however additional studies are certainly needed to add confidence to these findings before future clinical testing.

	All these valuable discussion points have been added to the revised manuscript, in both the results, discussion and conclusion sections, with additional reference citations included where appropriate.
5. The authors discussed difficulty of administering large solution volumes. What is the protein concentration in the solution administered? Discuss why the concentration couldn't be increased to enable smaller injection volume of the exact (neat) formulation? What is the highest protein concentration achieved in the PFNA/PFH combo?	5. In our experiments, we prepared fluoros protein samples with an intended final concentration of 10 μM for BSA, GFP, and Hb, 5 μM for IgG, and 1 μM for $\beta\text{-Gal}$. Dispersion efficiency results shown in Fig. 1c, indicate between 65 – 100% of the protein is dispersed into the fluoros solvent, depending on protein identity. Overall, this produced final protein concentrations ranging from 0.65 – 10 μM. As discussed in comment #4 above, attempts to reduce the injection volume to circumvent injection volume concerns resulted in too low of a delivered protein concentration to be detected by our substrate conversion assay. However, early evidence from studies using our 2nd generation dispersants suggests these improved molecules will address this limitation and enable significantly higher dispersed protein concentrations. It is our plan to publish these new molecules, and the extensive design process that went into their development, in a follow up manuscript when ready. Consequently, we feel this on-going work is beyond the scope of this initial publication demonstrating the feasibility of our platform technology.
6. Protein dispersion – this procedure starts with lyophilized protein – in contrast to the claims made by authors that this technology is an alternative to lyophilization etc. Discuss how this method would work with standard protein production processes that result in aqueous protein formulations.	6. We now include additional discussion in the conclusions section of the revised manuscript to describe use of this technology on concentrated protein fractions. In brief, we observed that use of lyophilized starting material yielded higher dispersion efficiencies relative to concentrated aqueous samples. Our added text now clarifies this observation.
7. Data in figure 2/3 versus their previous report (reference 14) that PFNA induces non-native secondary structure. Discuss the state of structure in the non-aqueous formulation (i.e., the actual formulation to be administered). It is important to understand the protein structure in the non-	7. During initial experiments we attempted to collect CD spectra directly from the PFOc protein samples to probe structure in the fluoros solvent. However, unfortunately, the solvent caused very high background scattering and our signal-to-noise ratio was poor; too low to make confident conclusions

aqueous drug product formulation as well as what happens to the protein once it is administered.	about protein structure. We therefore extracted the protein into buffer after heating of the PFOc samples to collect CD spectra probing thermal denaturation. This was the method to produce the data shown in Fig. 2a, which demonstrates that PFOc dispersed proteins remain structured up to 90°C. Importantly, the buffer extraction step used in these experiments serves to model the dissolution of PFOc dispersed proteins into physiologic solutions, like blood. As a result, our studies suggest that fluorour dispersed proteins will remain structured when intravenously administered.
8. The characterization assays are not performed on the representative formulations, and the starting point of samples in each assay uses different compositions/preps. Discuss the gap of not understanding structure of the proteins in the intended therapeutic formulation (to be administered as injected/inhaled as depicted in Fig. 1).	8. As requested, we have revised the text in various regions of the manuscript to articulate the nature of the samples being analyzed more clearly. For clarity, except for CD and animal studies, all other experiments tested the performance of fluorour-dispersed proteins on the representative formulation. In some cases the analytical assay performed required an aqueous environment (e.g., substrate conversion assay) and so we had to extract samples into water. However, the thermal, chemical, or biologic manipulation was done on the fluorour test article.
9. How complete is the extraction of protein into PBS from PFOc? If not 100%, discuss if only the fraction of active/native structure is extracted in PBS, leaving other fractions behind, and as a result skewing the relative activity assay data.	9. Extraction of proteins from PFOc into PBS was quantitative (100%). This was confirmed by Bradford assay, and is now discussed in the revised manuscript.
10. Without addressing these experimental gaps and gathering adequate data (at the minimum, discussing the gaps), it is highly speculative conclusion that they delivered “a first-in-class fluorochemical formulation paradigm that may limit, or altogether eliminate, the need for cold chain logistics of protein reagents and biopharmaceuticals.”	10. We thank the referee for their in-depth discussion and valuable suggestions. The revisions made as a result of these comments we feel significantly strengthen the manuscript overall. Based on the concerns raised, we have further revised the passage quoted in the referee’s comment to tone down the language of innovation and performance.

Reviewer #1 Thank you for the hard work on revisions to the manuscript. I just have two important technical questions and a couple grammatical issues to address.	Author Response to Reviewer #1 We thank the reviewer for their additional review of our manuscript. We have completed an additional NMR experiment to support the presence of a solvation shell, as well as addressed the additional editorial concerns, as described in more detail below.
1. The authors describe that the PFNA ligand “promiscuously adsorbs to protein surfaces”. It would be most helpful to describe this in terms of stoichiometry. I would have expected the interaction to be more like hydrophobic ion pairing rather than simple adsorption to the surface via hydrogen bonding. A description of the amount of PFNA that is bound is an important detail to convey.	1. We apologize for not including this information in our original manuscript. NMR studies performed during our early exploration of PFNA-protein interactions identified an average stoichiometry of 1,731 PFNA complexed per dispersed protein (ref: 14,15). For clarity, these prior studies focused on understanding the protein dispersive capabilities of PFNA, and did not report the thermal stabilization properties of this compound (which is the focus of this work). This additional discussion on stoichiometry, and appropriate references, have now been added to the revised manuscript.
2. The authors state that the proteins maintain a “solvation shell at the protein surface to avoid irreversible aggregation”. Can you please provide evidence (spectroscopic, water content, etc.) that this is true? Indirectly, it appears to be correct based on the structural assessment of the proteins, but this is a critical point and strong statement and it needs to be justified, especially since they also talk about water removal in these systems.	2. We now include ¹⁹F NMR results in Supplementary Figure 6 identifying a concentration-dependent chemical shift in the PFOc solvent’s -CF₃ peaks in the presence of dispersed β-Gal protein. The ¹⁹F signal for this experiment was referenced to a 2-(Trifluoromethyl)acrylic acid solution in D₂O contained within a coaxial insert. This data is reproduced below for convenience, and is the strongest evidence we have supporting a fluorinated solvent shell at the surface of the dispersed protein. While other techniques, including FTIR and vapor analysis, were considered, several technical limitations were identified that prevented us from utilizing these approaches. Nevertheless, this new NMR data shows direct interactions between the protein and PFOc solvent, and we believe now strongly supports our assertion of a perfluorocarbon solvation shell at the surface of dispersed proteins.

Supplementary Fig. 6: Stacked ¹⁹F NMR spectra demonstrating PFOc's -CF₃ chemical shift ($\Delta\delta$) as a function of increasing concentration of PFNA-dispersed β -Gal (0 – 100 μ M). Dashed lines are shown to aid in visualization of peak shift; PFNA concentration was kept constant at 1mM for all conditions.

3. In two places, sentences start with “To”. These should really start with “In order to...”

In the last paragraph, “it’s” should be “its”

3. We thank the referee for catching these typos and have made the appropriate corrections in the revised manuscript.

Reviewer #2

The authors have tried to diligently address the comments raised by this reviewer. Nevertheless, the section entitled 'In vivo equivalency and toxicity' is still substandard. It is highly recommended that the authors consult with a scientist with extensive pharmacokinetic experience and expertise to bring this section to a commonly acceptable level.

Besides general improvements, the following critical issues still need to be addressed:

1. In line 196, the authors claim that they want to detect differences in bioavailability. This is done in pharmacokinetics by comparing area-under-the-plasma-concentration time curves (AUC). Comparisons of half-life are utterly insufficient (and blatantly wrong) to achieve this goal.

Author Response to Reviewer #2

We thank the reviewer for their additional comments. During revision, we consulted with experts in PK/PD analyses to enhance the rigor of our interpretation and language in this section. Based on the reviewers' suggestions, and recommendations from the consulted PK experts, we performed several additional experiments to establish area-under-the-plasma-concentration time curves (AUC) for our protein samples. These are described in the revised manuscript, and detailed below.

1. To calculate protein AUC, as requested, we completed two additional *in vivo* experiments; the first to assay protein serum bioactivity at 5 and 15 minutes after injection to fill in the gaps in our previous curves. In a second, parallel experiment, we spiked freshly isolated mouse serum with varying amounts of protein to establish a calibration curve relating protein concentration to converted substrate fluorescence. Please see revised Figure 5a for the updated data, which has been reproduced below for convenience.

Together, this new analysis allowed us to calculate an AUC of 14.5 ± 4.3 mg/L*hr for β -Gal_{Saline} and 9.9 ± 3.1 mg/L*hr for β -Gal_{Ext. PFOc}. These AUC results were found to be statistically similar ($p = 0.09$; unpaired t-test). We believe this revised data supports our assertion that fluorour dispersion does not change the enzymatic function or pharmacokinetic properties of the complexed protein.

Fig. 5: *In vivo* half-life and safety of fluoros protein formulations. **a**, Time-dependent serum concentration of β -Gal delivered systemically in either saline (β -Gal_{Saline}, blue) or extracted PFOc (β -Gal_{Ext. PFOc}, green) vehicle. Data shown as average \pm s.d. of $n = 4$ -5 technical replicates. Statistical significance determined using Student's t-test and represented as n.s. = not significant, * $p < 0.05$. Area under the curve (AUC) determined in GraphPad Prism. **b**, Serologic toxicology results from C57BL/6J mice 24 hours after administration of saline (control), β -Gal_{Saline} or β -Gal_{Ext. PFOc}. Data shown as box and whisker plot \pm s.d. of $n = 4$ -5 technical replicates. Statistical significance determined using Student's t-test and represented as * $p < 0.05$; all other comparisons were found not to be significant ($p > 0.05$). Full size image and tabulated results can be found in Supplementary Fig. 9 and Supplementary Table S1, respectively. **c**, Representative histopathologic images of lung, kidney, liver, and spleen tissue section from C57BL/6J mice 24 hours after administration of saline (control), β -Gal_{Saline} or β -Gal_{Ext. PFOc}. Each imaging group consisted of $n = 4$ mice, with 4 random fields per section collected at 10X magnification in a blinded manner. Scale bar = 100 μ m. Full size image can be found in Supplementary Fig. 10.

2. In line 199, the authors claim that they monitor 'serum bioavailability over time'. Again, an utterly incorrect pharmacokinetic concept. What they likely do is monitor serum concentration levels (or enzymatic activity levels) over time.

2. We thank the reviewer for providing this important clarification. We have revised the text accordingly, and now indicate that we monitor the "time-dependent serum concentration of coated proteins..."

3. Line 200-203: The need for the extraction procedure still remains to be explained. The authors did that in the response to the reviewer comments, but not in the manuscript.	3. We thank the referee for catching this oversight, and have modified the manuscript text accordingly to provide additional details on the need for the pre-extraction step.
4. Line 205-208: While highlighting the lack of substantial differences in half-life, this is utterly insufficient for a pharmacokinetic comparison. Other pharmacokinetic parameters need to be compared as well (AUC, CL, Vd etc.)	4. As requested, and described above, we now include AUC data and compare the results for the β-Gal_{Ext.} PFOc and β-Gal_{Saline}, formulations. Respectfully, we argue that determining the additional PK parameters listed is outside the scope of this current manuscript, which is intended to establish proof-of-principle for the described formulation strategy. That said, we plan to conduct an in depth and rigorous PK study on our 2nd generation dispersants, given their ability to solubilize higher protein concentrations in the PFOc solvent relative to PFNA (the focus on this work). These follow up studies on the 2nd generation dispersants, however, are ongoing and will be separately published in due course.
5. Line 208-209: The authors claim that ‘Half-life is measured in our studies via a fluorescent substrate conversion assay...’. Again, this is scientific nonsense: Functional enzyme activity as a surrogate for active protein concentration is measured. Half-life is derived from those enzyme activity assay measurements over time via some type of regression analysis. This imprecision of description and wording of the authors in their scientific methodologies is disturbing, as it is likely not only limited to the pharmacokinetic assessments.	5. We recognize the reviewer’s concerns and appreciate the rigor of the feedback provided. To address the issue raised, we have modified the relevant text in our revised manuscript to read: “...It is important to note that half-life is calculated in our studies via regression analysis of results from a fluorescent substrate conversion assay, with subsequent translation of this bioactivity readout to protein concentration via a calibration curve (see methods and Supplementary Figure 8). Interpreting these results, area under the plasma concentration-time curves (AUC) were calculated as 14.5 ± 4.3 mg/L*hr for β-Gal_{Saline} and 9.9 ± 3.1 mg/L*hr for β-Gal_{Ext.} PFOc, and found to be statistically similar ($p = 0.09$; unpaired t-test)...” We also apologize for any perceived imprecision in our descriptions and wording. We have now carefully reviewed our description of protocols and procedures to ensure accuracy and uniformity of all methodologic explanations.

6. The toxicology assessment in lines 214-218 is largely meaningless as it is done with the extracted protein rather than the PFOc dispersed protein. As the latter is the goal to be administered to animals and humans, toxicity assessments should be focused on those formulations to assess their feasibility for translation into humans.

6. We believe that, because the current study shows proof of principle for thermally stabilized proteins using fluorous media, and that the described formulation is not the final product to be translated into humans, further toxicology studies are outside of the scope of this current work. That said, we do plan comprehensive toxicology studies on the final fluorous product, utilizing our second-generation dispersants (to be described in a separate manuscript), soon.

Reviewer #3

The revised article, 460388_1, Heat Stable and Intrinsically Sterile Liquid Protein Formulations, by Lawanprasert and coworkers provided adequate responses to most queries including additional data (thank you). However, two concerns remain (see below) regarding the practical utility of the technology/formulation presented here. But this reviewer agrees that the technology/process/formulation presented here serve as proof of principles for improved thermal stability and shelf sterility, acknowledging the lack of demonstration on bioequivalency, toxicity, administration of adequate dose etc. For example, with the possibility of improvements in protein partitioning and solubility, as noted by the authors in their response, one can imagine the dose regimen getting closer to reality (at least for some proteins).

Author Response to Reviewer #3

We thank the referee for their additional review our manuscript, and the feedback provided.

1. Extraction of the therapeutic

Extraction of the therapeutic by end user prior to administration is a high hurdle for injectables (that requires aseptic handling, preparation including transfer, and administration – the authors can review recent literature on these topics for challenges and controversies, as well as difficulty of implementing compendial and regulatory guidelines in addition to State-specific policies). Adding such a step defeats the purpose of improving access to drugs. The authors also noted that their process works better for lyophilized protein than aqueous formulation (aqueous formulation is the form produced for nearly all protein-based injectables, prior to converting to drug product). Thinking of the entire process for protein PFOc, (a) addition of lyophilization process, (b) addition of 1 or more custom devices for extraction/transfer/administration, and (c) the need for conducting the extraction by the end user, collectively makes the manufacturing cost and logistics high barriers. It should be noted that a lyophilized

1. We very much appreciate the referee’s insights, and feel that the importance of this topic necessitates additional discussion in the text. Consequently, we have added discussion in the concluding paragraphs that read:

“...Yet, further development of this formulation paradigm is necessary to reduce the barriers to its practical implementation. Of foremost priority is to maximize the soluble concentration of protein within the fluoros phase to enable direct injection of the therapeutic without compendial extraction or processing. Additionally, it would be beneficial to develop methods to disperse aqueous protein samples into the stabilizing fluoros media, rather than relying on lyophilized products. Such a goal may be realized by developing amphiphilic dispersants that undergo hierarchical assembly at the water-fluorous interface, thereby creating nanoscale receptacles that bind to proteins and mediate an aqueous to perfluorocarbon exchange.

These advances may be realized by building upon prior work in non-aqueous protein technologies. For example, our

drug product already provides vastly improved thermal stability including long-term shelf stability relative to aqueous formulations (as well as slowing down potential microbial growth). The authors are advised to consult, if needed, with biotech professionals who have experience in commercializing biologics drugs in global markets.

formulation approach is a contemporary alternative to hydrophobic ion pairing (HIP) methods reported several decades ago for protein dissolution in organic solvents..."

2. Lack of toxicity & bioequivalency data, lack of ability to produce adequate dose. The additional discussions on safety (lack of toxicity) and bioactivity/half-life are appreciated, however the lack of relevant experiments/data, as noted by this and other reviewers remains a gap. This is acknowledged by the authors in the revised manuscript.

A new gap emerged. With the added information, the achievable protein concentration appears to be orders of magnitude lower than the therapeutic IgG human doses (typically hundreds of mg total dose, and IgG concentration often exceeding 50 or 100 mg/mL in the recent years). So, the delivered dose via this new formulation will have to improve vastly, and ideally the process starting with aqueous formulation instead of requiring to lyophilize first. The authors are encouraged to add a few lines of discussion on why the presence of water (aq formulation) makes partitioning inefficient/inadequate, and if there is a way to circumvent it (without having to lyophilize).

2. Based on this concern about aqueous extraction, and our current reliance on lyophilized products, we have included relevant discussion in the revised concluding paragraphs, as detailed in our response above. It is our hope that this additional analysis now more clearly articulates the caveats of our platform, and highlights the areas of potential future development.

Reviewer #1 The authors have made great efforts to improve the quality of the manuscript, which I appreciate. I think the paper is ready to be published except for one issue.	Author Response to Reviewer #1 We thank the reviewer for their time and effort committed to improving our manuscript. We have now included additional revisions to correct the oversight recognized by the reviewer, as discussed below.
1. On lines 61-64, the authors claim that the stoichiometry is 1731. I find this to be strange. First, one should speak to a molar ratio (for example, 1731:1). First, how do determine this value to four significant figures? Second, this will vary from protein. Third, this value is not supported by the two cited references. They speak to systems where the ratios are 1000:1 to 1300:1. This needs to be corrected.	1. We apologize for this error and thank the reviewer for catching this oversight. The stoichiometry of 1731:1 was measured specifically for PFNA:GFP via ¹⁹F NMR experiments (ref. 15). The referee is correct that the stoichiometry varies depending on protein identity. We have now revised this passage in the main text to the following: “...Our prior work showed that perfluorononanoic acid (PFNA) adsorbs to proteins via hydrogen bonding with solvent accessible backbone moieties and amino acid side chains (Fig. 1b), with an average PFNA:protein stoichiometry that varies from 1000:1 to ~1700:1 depending on protein identity^{14,15}. Interestingly, we observed PFNA-mediated conformational changes to decorated proteins even at sub-stoichiometric ratios, suggesting that PFNA may alter protein PFC solubility through multiligand ensemble effects rather than a de-facto 1:1 protein-ligand interaction.¹⁴” As part of this discussion, we felt it important to include findings from our recently published work suggesting that PFNA may mediate protein dispersion into PFCs via ensemble (e.g. steric crowding) effects, rather than a canonical protein-ligand interaction. These revisions have been highlighted in the updated main text.

Reviewer #3

Thanks for trying to address my queries. I have no additional comments.

Author Response to Reviewer #3

We thank the reviewer for their constructive feedback during prior rounds of revision.

Reviewer #4

I have now assessed the response to Reviewer 2 and read the in vivo animal experimental description, results and interpretation.

Although this is quite interesting method, there are multiple limitations and i would list the things that needs to be addressed to improve the presentation and increase the validity of the in vivo studies:

1. Type of b-Gal with its molecular weight needs to be indicated

2. The dose given needs to be presented also in mg/kg to make the comparison to other studies feasible

Author Response to Reviewer #4

We greatly appreciate the referee's additional review our manuscript, and the feedback provided on the PK studies and analysis. We have conducted additional analyses and experiments to address the concerns raised by the reviewer, as discussed below. Importantly, PK calculations and analyses were conducted in collaboration with pharmacokinetic experts Drs. Arun Sharma and Asif Raza (Department of Pharmacology, Pennsylvania State University).

1. This information has now been added to the materials section found in the supplementary information.

2. Dose values (*D*) are now reported in Figure S9 of the supplementary information, which has been reproduced below for convenience.

Supplementary Fig. 9: (a,b) Log concentration of β-Gal in mouse serum versus sampling time for (a) β-Gal_{Saline} or (b) β-Gal_{Ext. PFOc}. Dashed line represents linear regime used for pharmacokinetic calculations. (c) Pharmacokinetic parameters for β-Gal_{Saline} or β-Gal_{Ext. PFOc} formulations, shown as mean value ± standard deviation. P values determined using Student's t-test.

3. Exact blood sampling times need to be investigated.

3. We now report exact sampling times in the methods section as 1, 5, 15, 30, 90 and 180 minutes post administration

4. The precision and accuracy data on the bioanalytical method needs to be presented. This is the key for any PK study

4. PK parameters are now represented in Supplementary Figure 9c as mean \pm standard deviation (see above). Statistical significance between formulations was determined using Student's t-test, with $p > 0.05$ considered not statistically significant.

We have also included the limit of detection for our *in vivo* serum half-life studies (Fig. 5a, reproduced below for convenience).

Fig. 5: Preliminary *in vivo* pharmacokinetics and acute toxicity. **a**, Time-dependent serum concentration of β -Gal delivered systemically in either saline (β -Gal_{Saline}, blue) or extracted PFOc (β -Gal_{Ext. PFOc}, green) vehicle. Data shown as average \pm s.d. of $n = 4-5$ technical replicates. Statistical significance determined using Student's t-test and represented as n.s. = not significant, * $p < 0.05$. Limit of detection (L.O.D.) is 0.66 mg/L (Supplementary Fig. 8, represented as dashed line on plot). Calculated pharmacokinetic parameters reported in Supplementary Fig. 9. **b**, Serologic toxicology results from C57BL/6J mice 24 hours after administration of saline (control), β -Gal_{Saline} or β -Gal_{Ext. PFOc}. Data shown as box and whisker plot \pm s.d. of $n = 4-5$ technical replicates. Statistical significance determined using Student's t-test and represented as * $p < 0.05$; all other comparisons were found not to be significant ($p > 0.05$). Full size image and tabulated results can be found in Supplementary Fig. 11 and Supplementary Table S1, respectively. **c**, Representative histopathologic images of lung, kidney, liver, and spleen tissue section from C57BL/6J mice 24 hours after administration of saline (control), β -Gal_{Saline} or β -Gal_{Ext. PFOc}. Each imaging group consisted of $n = 4$ mice, with 4 random fields per section collected at 10X magnification in a blinded manner. Scale bar = 100 μ m. Full size image can be found in Supplementary Fig. 12.

5. Dedicated PK analysis should be performed with reporting key systemic PK parameters (Cl, Vd) as it was already indicated by Reviewer 2. This is not difficult and would enhance the work. AUC calculation only by GraphPadPrism sounds a bit strange...	5. As requested, we have now calculated key PK parameters and reported these findings in Supplementary Figure 9 (see above). During revision, we also recalculated AUC values using the conventional trapezoidal rule calculations.
6. Fig. 5: In vivo half-life and safety of fluorous protein formulations title needs to be changed	6. We have renamed the title for Fig 5 to: Fig. 5: Preliminary in vivo pharmacokinetics and acute toxicity.
7. In Fig. 5 A should be presented in semilogarithmic scale as this is a the common way of presenting the data in PK.	7. As requested, we now show Fig. 5a as a semi-log plot. We have also added the limit of detection information in the caption to further clarify to the reader the precision of our analyses.
8. Ideally the stability of b-gal in blood, in vitro should be reported as well as it seems that the protein may undergo extensive systemic proteolysis which may impact also bioanalysis.	8. We thank the reviewer for this suggestion. We have now performed proteolysis studies for β-Gal in mouse serum, using the same substrate-conversion assay employed in our in vivo serum half-life studies. Results show that β-Gal remains active in mouse serum for >6 hours (Supplementary Fig. 10, reproduced below). This supports a conclusion that β-Gal's short in vivo serum half-life is due to rapid tissue distribution and/or renal elimination, rather than enzymatic degradation. This corroborates prior murine studies which found that β-Gal is cleared from serum in <15 minutes and distributes to vital tissues (refs. 20,21). This additional discussion has been added to the revised manuscript.

Supplementary Fig. 10: Fluorescence intensity (FI_{440nm}, relative fluorescence units) of the converted β -gal substrate, 4-Methylumbelliferyl- α -D-galactopyranoside, 5 minutes after its treatment with 1 μ M β -Gal pre-incubated in mouse serum at varying time points. Results shown as mean value \pm standard deviation.

9. It would be very valuable if authors could give a reference to other studies reporting the PK parameters for β -Gal, especially when presenting the $t_{1/2}$.

9. We now include two additional references reporting the serum half-life of β -Gal (refs. 20,21). In these reports, the protein was generally cleared from serum in <15 minutes (although half-life varied significantly based on the N-terminal residue), and was active in tissues for multiple hours. This discussion has been added to the revised manuscript.

10. It should be possible to address those questions rather easily. In addition, the authors should clearly highlight the limitations of existing studies and the need of further investigations.

10. As requested, we have now modified areas of the introduction and conclusion sections to further clarify the limitations of current approaches, and articulate the need for technologic innovation in protein formulation. This has been complemented by additional literature references. Particular passages in the revised manuscript discussing these aspects have been highlighted.